# MARS-Sep: Multimodal-Aligned Reinforced Sound Separation

**Zihan Zhang**[1]*, **Xize Cheng**[1]*, **Zhennan Jiang**[2], **Dongjie Fu**[1], **Jingyuan Chen**[1], **Zhou Zhao**[1], **Tao Jin**[1]†
[1]**Zhejiang University**    [2]**Institute of Automation, Chinese Academy of Sciences**
{zihanzhang.ai,jint_zju}@zju.edu.cn

## ABSTRACT

Universal sound separation faces a fundamental misalignment: models optimized for low-level signal metrics often produce semantically contaminated outputs, failing to suppress perceptually salient interference from acoustically similar sources. We introduce a preference alignment perspective, analogous to aligning LLMs with human intent. To address this, we introduce MARS-Sep, a reinforcement learning framework that reformulates separation as decision making. Instead of simply regressing ground-truth masks, MARS-Sep learns a factorized Beta mask policy that is steered by a preference reward model and optimized by a stable, clipped trust-region surrogate. The reward, derived from a progressively-aligned audio-text-vision encoder, directly incentivizes semantic consistency with query prompts. Extensive experiments on multiple benchmarks demonstrate consistent gains in Text-, Audio-, and Image-Queried separation, with notable improvements in signal metrics and semantic quality. Our code is available at https://github.com/mars-sep/MARS-Sep. Sound separation samples are available at https://mars-sep.github.io/.

## 1 INTRODUCTION

Sound separation (Liu et al., 2022; Dong et al., 2023; Cheng et al., 2025d; Chen et al., 2022; Mahmud & Marculescu, 2024; Ma et al., 2024; Huang et al., 2025b) is a foundational problem in audio processing with broad impact on downstream tasks such as speech recognition (Shi et al., 2022; Cheng et al., 2023; Kalda et al., 2024; Fu et al., 2024), spoken dialogue (Ji et al., 2024; Cheng et al., 2025b; Fu et al., 2025; Cheng et al., 2025a) sound event detection (Turpault et al., 2019; Yin et al., 2025), and acoustic scene analysis (Kim & Chang, 2024; Su et al., 2023). Beyond its standalone importance, separation also serves as a powerful data engine: by decomposing mixtures into isolated sources, it enables large-scale augmentation that improves robustness and generalization (Chiu et al., 2021; Yuan et al., 2022; Cheng et al., 2025c; Manilow et al., 2019). In this work, we focus on universal, query-conditioned separation—where the query can be audio, text, or image—and ask how to make the separated output not only signal-clean, but also semantically faithful to the user's intent.

Despite notable progress, prevailing methods are predominantly optimized for distortion or interference-based metrics including SDR, SIR, SAR (Vincent et al., 2006), SI-SDR (Roux et al., 2019) and give limited consideration to semantic alignment during training. This creates a *metric dilemma*: models optimized for waveform reconstruction can score high on signal-level metrics while leaving perceptually salient interference, thus violating semantic correspondence to the query.

Inspired by RLHF which learns a preference model from human data and uses it as a reward signal to steer the base LLM, we conceptualize query-conditioned sound separation as an identical **preference alignment problem**: the user query (audio, text, or image) is the preference, and the goal is to produce an output that maximizes semantic alignment with the query.

Based on this logic, we reformulate sound separation as a meta-reasoning task. We treat the base separation architecture as a base policy and use reinforcement learning as the optimization algo-

---

*Equal contribution.
†Corresponding author.

rithm. The human preference is captured by a learned multimodal reward model that provides a high-level semantic signal.

To this end, we propose **MARS-Sep**, a reinforcement learning framework that reformulates mask prediction as stochastic decision-making optimized with multimodal rewards. We cast mask generation as an actor-only trust-region optimization over a factorized Beta policy on time-frequency bins. Training uses a clipped surrogate with entropy regularization and normalized advantages, ensuring stable updates. Instead of focusing on low-level sampling details, our approach leverages multimodal rewards that holistically capture signal fidelity, interference suppression, and perceptual quality across audio, text, and visual queries. To provide reliable reward signals and mitigate reward hacking, we further introduce a progressive alignment strategy that fine-tunes the multimodal encoder to enhance cross-modal discrimination and stabilize policy learning.

We validate **MARS-Sep** on VGGSOUND-clean+ and MUSIC-clean+ (Dong et al., 2023) across Text-, Audio-, and Image-Queried separation. Extensive experiments show consistent gains over prior methods, improving SDR/SIR/SAR/SI-SDRi and notably higher CLAP score, while qualitative analyses highlight clearer suppression of non-target sources and better category discrimination.

Our contributions are summarized as follows:

- We formulate query-conditioned sound separation as a trust-region reinforcement learning problem, which optimizes a factorized Beta mask policy on time-frequency bins.

- We introduce a progressive alignment strategy that fine-tunes the multimodal encoder to enhance cross-modal discriminability and provide stable, informative reward signals, thereby mitigating reward hacking.

- We demonstrate consistent improvements across SDR/SIR/SAR and SI-SDRi, alongside higher CLAP scores and clearer qualitative separation for both synthesized and in-the-wild samples, confirming both signal-level and semantic gains.

## 2 RELATED WORK

### 2.1 UNIVERSAL AND QUERY-CONDITIONED SOUND SEPARATION

Research on isolating sources from complex mixtures has progressed from domain-specific settings—speech separation (Luo & Mesgarani, 2018; 2019; Zeghidour & Grangier, 2021; Subakan et al., 2021; Linhui et al., 2023) and music source separation (Luo et al., 2017; Rouard et al., 2023a; Luo & Yu, 2023)—toward Universal Sound Separation (USS) (Kavalerov et al., 2019; Wisdom et al., 2020), which aims to decompose arbitrary mixtures without class constraints. Key enablers include permutation invariant training (PIT) for resolving label permutations (Yu et al., 2017; Postolache et al., 2023) and mixture invariant training (MixIT) for leveraging unlabeled mixtures (Wisdom et al., 2020); large-scale resources such as AudioSet (Gemmeke et al., 2017) further catalyzed progress. Beyond audio-only models, integrating visual context (Majumder et al., 2021; Tan et al., 2023) or using class labels as queries (Chen et al., 2022; 2023) expands separation capability. In parallel, Query-Based Sound Extraction (QBSE) reframes the task as extracting user-specified content while suppressing irrelevant sources. By modality, label-queried systems are simple yet closed-set (Chen et al., 2022; Liu et al., 2024); text-queried approaches such as LASS-Net (Liu et al., 2022) enable open vocabulary but face joint-optimization and generalization hurdles; visual queries exploit images for grounding (Michelsanti et al., 2021; Gao & Grauman, 2021; Ye et al., 2024; Pian et al., 2024); and audio queries use exemplars to target abstract or indescribable sounds (Lee et al., 2019; Chen et al., 2022). Recent attempts unify modalities via cross-attention (Chen et al., 2023) or hybrid encoders (Rouard et al., 2023b), though joint training can limit generalization. Representative systems include CLIPSEP (Dong et al., 2022), which leverages visual data to improve text-queried training, and AudioSep (Liu et al., 2024), which couples CLAP (Wu et al., 2023) with a 14k-hour corpus to achieve strong zero-shot performance. Despite these advances, open-vocabulary robustness, multi-polarity operation (extraction and removal within one framework), and scalable multimodal composition (e.g., "dog barking in this image") remain challenging; query-mixup style training (Cheng et al., 2025d) offers a promising direction but still leaves room for improved semantic alignment and stability.

Beyond these discriminative and cross-modal systems, several recently proposed models extend query-conditioned separation into generative or flow-based paradigms. More recent generative approaches such as FlowSep (Yuan et al., 2025) employ rectified-flow matching to synthesize query-consistent sources directly in the latent space, and ZeroSep (Huang et al., 2025b) performs zero-training, text-conditioned separation using a pre-trained audio-language diffusion backbone. In contrast to fully synthesis-based models, DAVIS (Huang et al., 2024) and its successor DAVIS-Flow (Huang et al., 2025a) adopts a video-conditioned rectified-flow trajectory that remains time-aligned with the mixture, enabling evaluation under standard signal-level metrics. These developments collectively illustrate a shift toward richer multimodal conditioning and generative reasoning in sound separation, further motivating methods that unify signal fidelity with semantic alignment.

## 2.2 REINFORCEMENT LEARNING FOR LARGE LANGUAGE MODELS AND MULTIMODAL LEARNING

Reinforcement learning from human feedback (RLHF) has become a central paradigm for aligning large language models with human preferences. Early work (Ziegler et al., 2019; Ouyang et al., 2022) established the practice of training a reward model from pairwise preferences and optimizing the policy via Proximal Policy Optimization (PPO) (Schulman et al., 2017). Later methods reduced this pipeline's complexity, including Direct Preference Optimization (DPO) (Rafailov et al., 2023), which removes the explicit reward model, and more recent Group Relative Policy Optimization (GRPO) approaches (Guo et al., 2025; Zheng et al., 2025; Yu et al., 2025) that improve stability and reasoning over PPO-based RLHF.

These alignment strategies have also extended to multimodal models. Vision-R1 (Huang et al., 2025c) and R1-VL (Zhang et al., 2025a) apply structured rewards to improve grounding and chain-of-thought reasoning, while R1-reward (Zhang et al., 2025b) strengthens multimodal reward modeling to enhance semantic fidelity and robustness. Together, these works highlight the applicability of RLHF-style preference alignment beyond text-only LLMs.

## 3 METHOD

### 3.1 PRELIMINARIES

#### 3.1.1 UNIVERSAL SOUND SEPARATION

Universal Sound Separation (USS) is the task of isolating individual sound sources from an arbitrary audio mixture, without prior knowledge of the number or types of sources. Unlike domain-specific separation exemplified by speech enhancement or music source separation, USS aims to generalize across diverse acoustic conditions and sound categories.

Formally, let the observed mixture signal be denoted as $x(t) = \sum_{i=1}^{N} s_i(t)$, where $s_i(t)$ represents the waveform of the $i$-th underlying source and $N$ is the (unknown) number of sources. The goal of USS is to estimate a set of signals $\{\hat{s}_i(t)\}_{i=1}^{\hat{N}}$ such that each $\hat{s}_i(t)$ corresponds to one of the true sources $s_i(t)$, up to permutation and possibly scaling. That is, $x(t) \approx \sum_{i=1}^{\hat{N}} \hat{s}_i(t)$.

To achieve this, separation models typically operate in the time-frequency domain or directly on the waveform. Let $\mathbf{X} \in \mathbb{C}^{F \times T}$ denote the short-time Fourier transform (STFT) of the mixture, where $F$ and $T$ are the frequency and time dimensions. USS methods aim to construct masks $\{M_i \in [0,1]^{F \times T}\}$ such that $\hat{\mathbf{S}}_i = M_i \odot \mathbf{X}$ with $\odot$ denoting element-wise multiplication. The inverse STFT is then applied to obtain time-domain estimates $\hat{s}_i(t)$.

#### 3.1.2 OMNISEP: UNIFIED OMNI-MODALITY SOUND SEPARATION WITH QUERY-MIXUP

OmniSep (Cheng et al., 2025d) provides the base separation architecture: a frozen ImageBind (Girdhar et al., 2023) encoder maps audio/image/text inputs to a shared feature space and a Separate-Net (U-Net over STFT magnitudes) predicts masks. Let $Q_A$ be the audio query, $Q_V$ be the visual query, and $Q_T$ be the text query. Training in OmniSep mixes queries across modalities via Query-Mixup

$$Q = \frac{w_a Q_A + w_v Q_V + w_t Q_T}{w_a + w_v + w_t}, \quad w_a, w_v, w_t \in [0,1] \tag{1}$$

and combines intermediate masks into final masks $\hat{M}$ through channel-wise weighting; the supervised objective is the sum of weighted binary cross-entropy (WBCE) losses on ideal masks. OmniSep also supports negative queries by introducing a negative query weight $\alpha$ to adjust the query as $Q' = (1 + \alpha)Q - \alpha Q_N$ to remove interfering content, and, more broadly, frames omni-modal querying with a frozen ImageBind backbone within a unified audio/text/video-queried separation paradigm.

## 3.2 MARS-SEP: REINFORCEMENT LEARNING FOR MULTI-SOURCE UNIVERSAL SOUND SEPARATION ENHANCEMENT

As established in our introduction, we frame sound separation as a preference alignment problem, analogous to the RLHF pipeline for LLMs. To operationalize this framework (see Figure 1), we must define three core components:

**Base Policy ($\pi_\theta$).** The sound separation model that takes the state (mixture spectrogram $X$ and query $Q$) and produces an action (the mask $M$).

**Preference Reward Model ($R$).** The model that scores the alignment. We use our progressively-aligned multimodal encoder to provide this scalar reward, measuring semantic consistency between the separated audio and the query.

**Optimization Process.** The policy updating algorithm that steers the base policy to maximize the reward. We adapt a stable, trust-region policy optimization algorithm for this purpose.

During the training stage, the separator produces a mask prediction that parameterizes the *new policy* ($\alpha_{new}, \beta_{new}$), while a snapshot from the previous step provides the *old policy* ($\alpha_{old}, \beta_{old}$). Masks are sampled from the old policy to ensure stable training, and the separated audio is compared against audio/text/video queries in a shared embedding space using a fine-tuned ImageBind encoder with a multimodal fusion module. The cosine similarity between the separated audio embedding and the fused query embedding yields a scalar reward, which is normalized by a running baseline and group-relative scaling to form advantages. These are combined with the log-probability ratio between old and new policies (i.e., $\log\pi_{\text{old}}$ and $\log\pi_{\text{new}}$) under a clipped surrogate objective, supplemented by entropy regularization for exploration and a KL penalty for stability. The current policy is then snapshotted to serve as the old policy in the next iteration.

### 3.2.1 TRUST-REGION-STYLE POLICY OPTIMIZATION FOR STABLE MASK SAMPLING

We formulate query-conditioned sound separation as a standard Markov Decision Process (MDP) $(\mathcal{S}, \mathcal{A}, T, R, \gamma)$. The state space $\mathcal{S}$ consists of the mixture spectrogram $X$ and the query $Q$, while the action space $\mathcal{A}$ corresponds to masks $M$. The transition $T$ is deterministic, $\hat{y} = s(X, M)$, reconstructing the waveform. The reward function $R$ is defined by the similarity between the separated waveform and the multimodal query. Our goal is to train a policy $\pi_\theta(M \mid X, Q)$ that maximizes the cumulative designed reward. Let $X$ be the magnitude spectrogram of a mixture and $\theta$ the separator parameters that produce a deterministic mask proposal $P_\theta(X, Q) \in [0, 1]^{H \times W \times K}$, where $H$ is the number of frequency bins, $W$ is the number of time frames and $K$ denotes the number of target sources to be separated. We turn this proposal into a stochastic policy over masks by a factorized Beta distribution

$$\pi_\theta(M \mid X, Q) = \prod_{h,w,k} \text{Beta}\big(M_{h,w,k};\ \alpha_{h,w,k},\ \beta_{h,w,k}\big), \quad \alpha = 1 + \kappa P_\theta,\ \beta = 1 + \kappa(1 - P_\theta), \quad (2)$$

with concentration scale $\kappa > 0$. Reparameterized sampling $M \sim \pi_\theta(\cdot \mid X, Q)$ yields a mask that is close to the proposal yet retains exploration. The masked magnitude is combined with the mixture phase to reconstruct a waveform $\hat{y} = s(X, M)$; the ground-truth component $y^\star$ is reconstructed analogously from ideal masks (only for reward computation during training). The factorized Beta parametrization aligns with the $[0, 1]$ support of masks and offers a transparent exploration-exploitation knob via the concentration scale $\kappa$, which we anneal to avoid degenerate near-binary masks early in training.

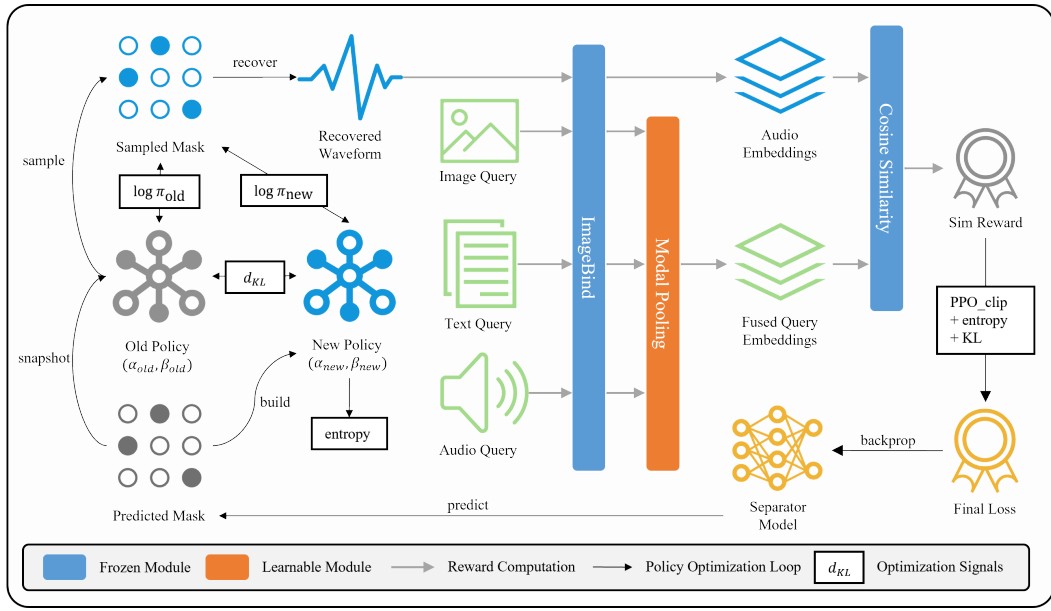

Figure 1: The reinforcement learning loop of MARS-Sep. The separator generates stochastic mask actions from a Beta-distributed policy, while a frozen snapshot serves as the old policy for stable optimization. Multimodal rewards derived from audio, text, and visual embeddings guide policy updates, with entropy and KL regularization enhancing exploration and stability.

At each training step we sample from a frozen old policy $\pi_{\theta_{\text{old}}}$ (a snapshot from the previous step), reconstruct the waveform $\hat{y}$ from the sampled mask $M$, and compute a scalar reward $R$ against the query-conditioned targets. A moving-average baseline $b$ yields the advantage $A = R - b$. To stabilize updates, we adopt a clipped trust-region style surrogate in the spirit of PPO, using GRPO of the advantage. Concretely, define the importance ratio

$$r_\theta(M) = \frac{\pi_\theta(M \mid X, Q)}{\pi_{\theta_{\text{old}}}(M \mid X, Q)} = \exp\big(\log \pi_\theta(M) - \log \pi_{\theta_{\text{old}}}(M)\big), \tag{3}$$

and let $\tilde{A} = \frac{A - \mu(A)}{\sigma(A) + \varepsilon}$ be the group-relative advantage. The clipped surrogate objective with entropy regularization and KL penalty is

$$\mathcal{J}_{\text{clip}}(\theta) = \mathbb{E}_{M \sim \pi_{\theta_{\text{old}}}}\Big[ \min\big(r_\theta(M)\,\tilde{A},\ \text{clip}(r_\theta(M), 1-\epsilon, 1+\epsilon)\,\tilde{A}\big) + \lambda_H\,\mathcal{H}\big(\pi_\theta\big) - \lambda_{\text{KL}}\,\text{KL}\big(\pi_\theta \parallel \pi_{\theta_{\text{old}}}\big)\Big], \tag{4}$$

and the loss minimized in training is $\mathcal{L}_{\text{RL}}(\theta) = -\mathcal{J}_{\text{clip}}(\theta)$. Here $\mathcal{H}$ denotes the entropy of the factorized Beta policy, $\epsilon$ is the clipping range, and $\lambda_H, \lambda_{\text{KL}} > 0$ control exploration and the trust region, respectively. In practice, $\log \pi$ factorizes over bins; we broadcast $\tilde{A}$ to the mask shape and estimate expectations with Monte Carlo samples per iteration. The old policy $\pi_{\theta_{\text{old}}}$ is updated to the current snapshot after each optimization step, yielding a single-epoch PPO update that preserves the original training loop while markedly improving stability.

This design preserves the benefits of reinforcement learning while avoiding the instability of plain policy gradients, leading to more reliable convergence for mask-based separation. Importantly, it achieves this without introducing additional value networks or complex estimators, keeping the optimization efficient while directly tying policy updates to multimodal reward signals.

### 3.2.2 MULTIMODAL REWARD

To optimize the separation policy, we define a reward that measures how well the separated audio waveform $\hat{y}$ semantically matches the target query across modalities in a unified embedding space

provided by ImageBind. We project audio waveforms, text queries, and sampled video frames into a shared space via ImageBind encoders $\phi_a(\cdot)$, $\phi_t(\cdot)$, and $\phi_v(\cdot)$, respectively.

All embeddings are $\ell_2$-normalized, and similarity is measured with cosine similarity:

$$\mathrm{sim}(u, v) = \left\langle \frac{u}{\|u\|}, \frac{v}{\|v\|} \right\rangle. \tag{5}$$

**Unimodal rewards.** Given separated audio $\hat{y}$, ground-truth audio $y^\star$, a text query embedding $t^\star$, and a video frames embedding $v^\star$, we compute:

$$r_{a \to a} = \mathrm{sim}\big(\phi_a(\hat{y}), \phi_a(y^\star)\big), \quad r_{t \to a} = \mathrm{sim}\big(\phi_a(\hat{y}), \phi_t(t^\star)\big), \quad r_{v \to a} = \mathrm{sim}\big(\phi_a(\hat{y}), \phi_v(v^\star)\big). \tag{6}$$

These terms measure acoustic fidelity, semantic alignment with text, and consistency with visual context, respectively.

**Aggregation strategy: Query-pooling.** Instead of comparing unimodal similarities separately, we fuse the target-side multimodal embeddings into a joint representation using Multi-Modal Low-Rank Bilinear Pooling (MLBP) ((Kim et al., 2017)) and compare it directly to the separated audio embedding. Specifically, $z^\star = \mathrm{MLBP}\big(\phi_a(y^\star), \phi_t(t^\star), \phi_v(v^\star)\big)$, and the scalar reward is $R = \mathrm{sim}\big(\phi_a(\hat{y}), z^\star\big)$. Detailed implementation of MLBP is described in Appendix A.

The motivation for pooling is to ensure the reward captures joint multimodal consistency rather than independent unimodal matches. Audio, text, and vision carry complementary cues: audio encodes acoustic details, text conveys semantic categories, and vision provides environmental context. If each is compared separately, the reward may overweight a single modality. By applying low-rank bilinear pooling, we explicitly model multiplicative interactions between modalities (e.g., a textual query specifying an instrument that also appears visually). This fused target anchor $z^\star$ encourages the separated audio to simultaneously align with all modalities, yielding more semantically faithful and robust rewards. This asymmetric design mirrors the implementation: the separated audio remains in its native representation while the target modalities are fused into a semantic anchor. This reduces variance from stochastic mask sampling and provides a stable training signal.

In the next section, we describe the progressive fine-tuning curriculum used to initialize this policy with robust cross-modal alignment before RL optimization.

## 3.3 MULTIMODAL ENCODER FINE-TUNING VIA PROGRESSIVE ALIGNMENT

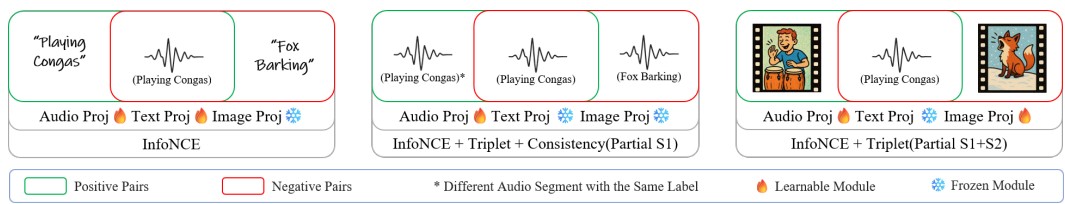

Figure 2: Progressive fine-tuning strategy for sound source discrimination and separation. Encoders remain frozen while task-specific heads are gradually unfrozen and each stage builds on the best checkpoint from the previous one. The two latter stages are trained with a fraction of the former aligned paired data to avoid catastrophic forgetting.

To enhance discrimination between same- and different-source signals, we apply a multimodal contrastive fine-tuning objective on ImageBind: for each audio segment, positives pair it with an audio clip, video frame, or label text of the same class, while negatives pair it with content from other classes; by optimizing a contrastive loss over these multimodal pairs, the model is encouraged to bring embeddings of semantically consistent sources closer together, while pushing apart those belonging to different sound categories.

The fine-tuning process, as shown in Figure 2, is organized into three sequential stages, each with a distinct training objective, and each stage begins from the best-performing checkpoint obtained in the previous one. This curriculum design allows the model to move gradually from semantic grounding to intra-class discrimination and finally to multimodal alignment.

### 3.3.1 SAMPLE PAIR CONSTRUCTION FOR CONTRASTIVE LEARNING

We build multimodal contrastive pairs with the audio clip as the anchor. Positives pair the anchor with (i) its label text, (ii) another audio instance from the same class, or (iii) frames from the temporally aligned video segment. Negatives use labels, audio, or frames from other classes or temporally mismatched segments. This unified scheme pulls matched embeddings together while pushing mismatched ones apart.

In the first stage, the model is trained to align audio signals with their corresponding textual labels. At this point, all modality encoders (audio, text, vision) and postprocessors are kept frozen to preserve their pretrained representations, while only the projection heads together with a shared temperature parameter are unfrozen and updated. The training objective is a symmetric InfoNCE loss, which encourages paired audio-text embeddings to converge while repelling mismatched pairs. Formally, for a batch of size $N$, with normalized embeddings $z_a^i$ and $z_t^i$, the loss is defined as

$$\mathcal{L}_{\text{S1}} = -\frac{1}{2N} \sum_{i=1}^{N} \left[ \log \frac{\exp(\langle z_a^i, z_t^i \rangle / \tau)}{\sum_{j=1}^{N} \exp(\langle z_a^i, z_t^j \rangle / \tau)} + \log \frac{\exp(\langle z_t^i, z_a^i \rangle / \tau)}{\sum_{j=1}^{N} \exp(\langle z_t^i, z_a^j \rangle / \tau)} \right], \tag{7}$$

where $\tau$ is a learnable temperature scaling factor. This stage establishes the initial semantic grounding of audio in the shared embedding space while limiting parameter updates to lightweight layers.

The second stage focuses on audio-audio discrimination. Again, all encoders and postprocessors remain frozen, while the audio projection head and shared temperature are unfrozen to adapt representations for finer discrimination. Given an anchor audio clip, another clip of the same class is selected as the positive, while a clip of a different class serves as the negative. The objective combines several terms: an InfoNCE loss to enforce alignment between same-class pairs, a triplet loss to guarantee a margin between positive and negative similarities, and a consistency loss to ensure invariance to perturbations. Specifically,

$$\mathcal{L}_{\text{S2}} = \lambda_1 \, \mathcal{L}_{\text{InfoNCE}}(z_1, z_2) + \lambda_2 \, \max(0, [1 - \cos(z_1, z_2)] - [1 - \cos(z_1, z_n)] + m) + \lambda_3 \|z_1 - z_2\|^2, \tag{8}$$

where $z_1$ and $z_2$ are embeddings of audio from the same class, $z_n$ is a negative sample, and $m$ is a margin hyperparameter. To mitigate catastrophic forgetting, a fraction of audio-text pairs from stage one are mixed into training, ensuring that semantic alignment is preserved while discrimination improves.

The third stage introduces visual grounding through audio-video pairs. All encoders and postprocessors remain frozen, but the audio and vision projection heads are unfrozen and trained jointly, along with the shared temperature. Uniformly sampled frames from the corresponding video provide the positive modality, while frames from other videos or temporally misaligned portions serve as negatives. The objective again includes an InfoNCE term between audio and video embeddings, along with a triplet loss that uses mismatched video frames as hard negatives. To maintain previously acquired capabilities, the objectives from stage one and stage two are partially incorporated. The overall loss can be expressed as

$$\mathcal{L}_{\text{S3}} = \mu_1 \, \mathcal{L}_{\text{InfoNCE}}(z_a, z_v^+) + \mu_2 \, \mathcal{L}_{\text{Triplet}}(z_a, z_v^+, z_v^-) + \mu_3 \, \mathcal{L}_{\text{S1}} + \mu_4 \, \mathcal{L}_{\text{S2}}, \tag{9}$$

where $z_v^+$ and $z_v^-$ are embeddings of positive and negative video samples, and the coefficients $\mu_i$ balance the relative contributions.

At the end of stage 1 and 2, the best checkpoint is used to initialize the subsequent stage. Stage one therefore provides a semantic anchor via audio-text alignment, stage two sharpens class discrimination through audio-audio comparison, and stage three consolidates multimodal grounding by linking

audio with vision. This progressive fine-tuning procedure, in which encoders are kept frozen and only task-relevant heads and scaling parameters are successively unfrozen, ensures that the model evolves in a stable and interpretable manner, acquiring increasingly sophisticated capabilities for sound source discrimination and separation.

## 4 EXPERIMENTS

### 4.1 EXPERIMENT SETUP

We evaluate our approach on two widely used audio-visual separation benchmarks, **VGGSound** (Chen et al., 2020) and **MUSIC** (Zhao et al., 2018). VGGSound is a large-scale dataset with over 300 sound categories collected from YouTube videos, offering substantial acoustic and visual diversity; MUSIC is a smaller dataset of solo and duet music performance videos spanning a variety of instruments, which emphasizes structured harmonic signals and thus provides a complementary and cross-domain setting. Training and data preprocessing details are provided in Appendix B.

We adopt standard separation metrics for evaluation, including Signal-to-Interference Ratio (SIR), Signal-to-Distortion Ratio (SDR), Signal-to-Artifact Ratio (SAR), and Scale-Invariant SDR improvement (SI-SDRi)[1].

We additionally report the **CLAP score**, which measures the semantic consistency between the separated audio and its textual label using a contrastive language-audio pretraining model. While traditional signal-level metrics evaluate separation quality in terms of distortion, interference suppression, and artifact reduction, the CLAP score complements them by capturing whether the separated waveform preserves the intended semantic content. Together, these metrics provide a comprehensive assessment of both perceptual signal fidelity and semantic correctness of the separation output.

We select five representative baselines for comparison. **CLIPSep-NIT** (Dong et al., 2023) (the noise-invariant training version released by the authors) employs CLIP embeddings to guide separation with either visual or textual cues. **AudioSep** (Liu et al., 2024) adopts large-scale training with text queries to achieve strong open-domain generalization. **OmniSep** (Cheng et al., 2025d) integrates multiple modalities into a unified separation framework, highlighting the potential of multimodal fusion. **LASS-Net** (Liu et al., 2022) addresses the language-queried audio source separation task via a joint Transformer-based query encoder and ResUNet separation network, conditioned on natural-language descriptions. Second, **iQuery** (Chen et al., 2023) formulates instruments as learnable audio-query prototypes and leverages visually-named cross-modal attention to disentangle and separate instrument sounds from videos.

In addition to the above, we further include two generative approach and one zero-shot approach for comparison. **FlowSep** (Yuan et al., 2025) introduces a generative rectified flow matching (RFM) model in the VAE latent space to synthesize separated audio from noise under language query guidance, thereby enabling cleaner outputs in highly overlapping scenarios. **ZeroSep** (Huang et al., 2025b) proposes a zero-training audio separation framework that repurposes pre-trained text-guided audio diffusion models to perform open-set language-queried separation without task-specific fine-tuning. We also include **DAVIS-Flow** (Huang et al., 2025a), a visually-guided generative audio-visual separation framework that applies flow-matching to synthesize separated spectrograms from noise.

### 4.2 MAIN RESULTS

We first present results on the **VGGSound-clean+** dataset, a refined subset of VGGSound that filters out noisy annotations and ensures higher-quality alignment between audio and visual streams. *MARS-Sep* demonstrates the strongest overall performance across text, audio, image, and composed queries (Table 1). It attains the highest SDR and CLAP score across modalities and achieves the best SI-SDRi in three settings (tied for best under audio queries). On a subset of measures, the balance shifts toward *OmniSep* with notably higher SIR for audio queries and higher SAR for image and composed queries, though the margins are modest. Taken together, these results indicate that

---

[1]A detailed description of the SI-SDR and SI-SDRi calculation procedure is presented in Appendix C.

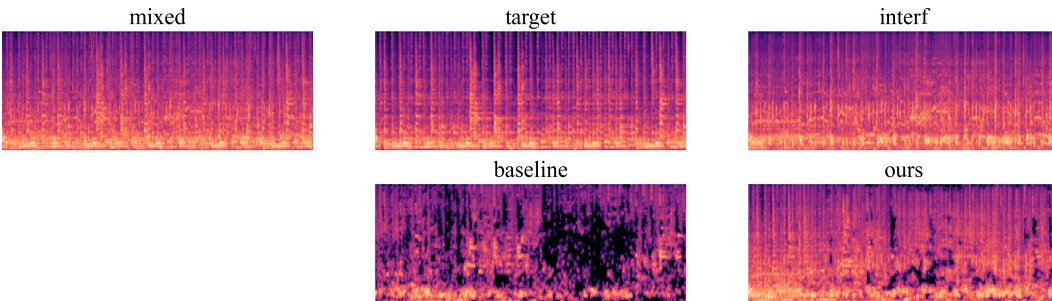

Figure 3: Log-mel spectrograms of separated audio from different query modalities on VGGSOUND-clean+ dataset. The target source is "cattle bovinae cowbell". From left to right: (a) Mixture of "cattle bovinae cowbell" and "tap dancing"; (b) Ground-truth "cattle bovinae cowbell"; (c) Interference "tap dancing"; (d) Separation with text query by the baseline model; (e) Separation with text query by our model.

reinforcement learning with multimodal rewards improves semantic fidelity and signal quality on balance, while remaining competitive or superior on the remaining metrics.

To further validate cross-domain generalizability, we evaluate on **MUSIC-clean+**, which is derived from the MUSIC dataset and focuses on solo and duet instrumental performances. Compared with VGGSound, MUSIC emphasizes structured harmonic and timbral patterns rather than broad acoustic diversity, making it a complementary benchmark. As reported in Table 2, *MARS-Sep* again achieves clear gains over competing approaches under all query modalities, especially notably higher CLAP scores than all baselines, highlighting that our separated signals preserve semantic consistency with the intended text source, confirming that our method not only handles open-domain separation but also excels in structured, music-centric scenarios.

We provide log-mel spectrograms of a representative sample from the test set of VGGSOUND in Figure 3 for result visualization. Compared with the baseline, MARS-Sep suppresses non-target components more selectively, preserving the target's harmonic ridges and temporal continuity instead of the blocky dropouts visible in the baseline spectrogram.

We additionally report CLAP scores for the generative separation frameworks FlowSep and ZeroSep, which do not produce mask-based or waveform-aligned outputs and are therefore not directly comparable under signal-level metrics. As shown in Table 3, these generative models exhibit substantially higher variance and often inflated CLAP similarity due to their synthesis-based decoding. In contrast, MarsSep provides far more stable and consistent semantic alignment, while also enabling evaluation under standard separation metrics. To further assess the quality of the separated audio, we introduce CLAP audio scores, which measure the semantic similarity between the separated audio and the target audio. This additional evaluation highlights MarsSep's ability to maintain strong semantic fidelity to the target, even when compared to models with synthesis-based outputs, where CLAP scores tend to be more volatile.

As for ablation studies, the impact of **hyperparameter settings, modality query embedding fusion module, progressive finetuning strategy, discrimination ability of multimodal encoder with or without finetuning** and other significant factors are discussed in Appendix E.1.

More samples including synthesized audio and in-the-wild audio are shown in Appendix E.6.

## 5 CONCLUSION

We present MARS-Sep, a multimodal-aligned reinforced sound separation approach that frames sound separation as stochastic decision-making guided by multimodal rewards, enforcing semantic consistency with audio, text, and visual queries rather than optimizing only signal-level metrics. Built on a trust-region style policy with progressive alignment of multimodal encoders, it achieves stable training and strong cross-modal discrimination. Experiments on VGGSOUND-clean+ and MUSIC-clean+ show consistent improvements in fidelity and semantic alignment, advancing semantically aware separation that better matches perceptual quality.

Table 1: Comparison of sound separation performance among different methods on **VGGSOUND-clean+** dataset. Metrics include SIR, SDR, SAR, and SI-SDRi (all in dB), and CLAP score (%).

| Methods | VGGSOUND-clean+ | | | | |
| | Mean SDR↑ | Mean SIR↑ | Mean SAR↑ | Mean SI-SDRi↑ | Mean CLAP$_t$ ↑ |
|---|---|---|---|---|---|
| *Text Query Sound Separation* | | | | | |
| LASS-Net (Liu et al., 2022) | 3.98±1.02 | 7.63±0.85 | 4.24±1.00 | 4.25±0.76 | 5.12±0.71 |
| CLIPSEP-NIT (Dong et al., 2023) | 2.71±0.87 | 4.58±1.37 | 13.60±0.68 | 2.41±0.53 | 7.97±0.94 |
| AudioSep (Liu et al., 2022) | 6.26±0.87 | 8.69±0.90 | 12.85±0.92 | 4.01±0.59 | 8.21±0.96 |
| DAVIS-Flow (Huang et al., 2025a) | 6.60±1.02 | 8.99±0.93 | 13.48±0.85 | 4.32±1.03 | 6.57±0.94 |
| OmniSep (Cheng et al., 2025d) | 6.70±0.66 | 9.04±0.98 | 13.61±0.77 | 4.38±0.48 | 8.98±0.89 |
| MARS-Sep (ours) | **6.91±0.68** | **9.14±1.00** | **13.73±0.77** | **4.55±0.44** | **9.03±0.94** |
| *Audio Query Sound Separation* | | | | | |
| OmniSep (Cheng et al., 2025d) | 7.15±0.65 | **11.65±1.02** | 11.84±0.81 | 4.35±0.52 | 8.60±0.91 |
| MARS-Sep (ours) | **7.33±0.67** | 11.63±1.00 | **12.00±0.84** | **4.36±0.50** | **8.91±0.91** |
| *Image Query Sound Separation* | | | | | |
| CLIPSEP-NIT (Dong et al., 2023) | 4.61±0.82 | 8.11±1.32 | 12.06±0.78 | 3.48±0.60 | 8.50±0.92 |
| iQuery (Chen et al., 2023) | 6.20±0.78 | 9.59±0.88 | 13.45±1.01 | 3.77±0.46 | 6.08±1.12 |
| DAVIS-Flow (Huang et al., 2025a) | 6.52±1.01 | 9.87±0.98 | 13.54±0.93 | 4.32±0.96 | 8.89±1.02 |
| OmniSep (Cheng et al., 2025d) | 6.66±0.65 | 10.00±1.05 | **13.73±0.76** | 4.43±0.50 | 8.79±0.89 |
| MARS-Sep (ours) | **6.93±0.67** | **10.18±1.04** | 13.41±0.72 | **4.57±0.47** | **9.19±0.91** |
| *Composed Omni-modal Query Sound Separation* | | | | | |
| OmniSep (Cheng et al., 2025d) | 7.79±0.72 | **10.76±1.00** | **14.53±0.93** | 5.16±0.47 | 8.85±0.92 |
| MARS-Sep (ours) | **7.93±0.75** | 10.65±1.00 | 14.49±0.95 | **5.20±0.45** | **9.22±0.90** |

Table 2: Comparison of sound separation performance among different methods on **MUSIC-clean+** dataset.

| Methods | MUSIC-clean+ | | | | |
| | Mean SDR↑ | Mean SIR↑ | Mean SAR↑ | Mean SI-SDRi↑ | Mean CLAP$_t$ ↑ |
|---|---|---|---|---|---|
| *Text Query Sound Separation* | | | | | |
| LASS-Net (Liu et al., 2022) | 9.98±0.99 | 14.63±1.17 | 12.24±1.10 | 8.99±0.76 | 4.92±0.71 |
| CLIPSEP-NIT (Dong et al., 2023) | 11.03±0.98 | 16.40±1.38 | 17.37±0.97 | 7.53±0.90 | 5.29±0.96 |
| AudioSep (Liu et al., 2022) | 11.23±0.92 | 16.90±1.31 | 17.29±0.90 | 8.56±0.84 | 5.48±1.02 |
| DAVIS-Flow (Huang et al., 2025a) | 8.97±0.85 | 15.99±1.55 | 17.88±0.95 | 9.23±1.03 | 5.53±0.75 |
| OmniSep (Cheng et al., 2025d) | 12.37±0.85 | 17.51±1.16 | 17.96±0.90 | 9.18±0.79 | 5.41±0.98 |
| MARS-Sep (ours) | **12.91±0.93** | **17.61±1.17** | **18.28±0.93** | **9.85±0.82** | **6.18±0.93** |
| *Audio Query Sound Separation* | | | | | |
| OmniSep (Cheng et al., 2025d) | 10.37±0.86 | 17.76±1.05 | 14.51±0.88 | 7.18±1.07 | 5.39±1.01 |
| MARS-Sep (ours) | **11.73±0.88** | **19.65±1.14** | **15.25±0.86** | **8.38±1.03** | **5.64±1.06** |
| *Image Query Sound Separation* | | | | | |
| CLIPSEP-NIT (Dong et al., 2023) | 11.64±0.98 | 18.40±1.26 | 17.04±1.05 | 8.27±0.94 | 5.97±0.94 |
| i-Query ((Chen et al., 2023)) | 12.52±0.84 | 17.05±1.01 | 17.65±1.23 | 9.44±0.99 | 5.50±1.22 |
| DAVIS-Flow (Huang et al., 2025a) | 13.52±1.14 | 19.00±1.02 | 17.54±0.93 | 10.32±0.85 | 5.99±1.00 |
| OmniSep (Cheng et al., 2025d) | 13.03±0.96 | 18.97±1.16 | 17.88±1.00 | 10.21±0.89 | 6.53±1.03 |
| MARS-Sep (ours) | **13.64±1.06** | **19.24±1.16** | **18.05±1.06** | **10.70±0.89** | **6.94±1.06** |
| *Composed Omni-modal Query Sound Separation* | | | | | |
| OmniSep (Cheng et al., 2025d) | 13.29±0.96 | 19.55±1.17 | 17.88±0.96 | 10.22±0.89 | 6.35±1.05 |
| MARS-Sep (ours) | **13.89±0.98** | **19.90±1.18** | 17.99±0.97 | **10.78±0.81** | **6.82±0.99** |

Table 3: CLAP score comparison (Text-queried) across generative separation frameworks on the MUSIC-clean+ and VGGSOUND-clean+ datasets.

| Method | Dataset | $CLAP_t$ score (%) | $CLAP_a$ score (%) |
|---|---|---|---|
| ZeroSep | | $20.02 \pm 15.14$ | $22.86 \pm 18.55$ |
| FlowSep | MUSIC-clean+ | $10.67 \pm 14.17$ | $39.25 \pm 29.86$ |
| MarsSep (Ours) | | $6.18 \pm 0.93$ | $21.56 \pm 1.08$ |
| ZeroSep | | $15.91 \pm 14.17$ | $22.65 \pm 19.98$ |
| FlowSep | VGGSOUND-clean+ | $8.84 \pm 13.27$ | $56.07 \pm 19.57$ |
| MarsSep (Ours) | | $9.03 \pm 0.94$ | $18.70 \pm 1.23$ |

ACKNOWLEDGMENTS

This work was supported by the National Natural Science Foundation of China under Grant No. U25B2064, the "Pioneer" and "Leading Goose" R&D Program of Zhejiang under (Grant No. 2025C02110), Public Welfare Research Program of Ningbo under (Grant No. 2024S062), and Yongjiang Talent Project of Ningbo under (Grant No. 2024A-161-G).

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

## A   MULTI-MODAL LOW-RANK BILINEAR POOLING (MLBP).

Based on (Kim et al., 2017), let there be $K$ modalities with input vectors $x^{(k)} \in \mathbb{R}^{d_k}$. Each is projected to a shared dimension $d$ via linear transformations without bias:

$$\tilde{x}^{(k)} = W_k x^{(k)}, \quad W_k \in \mathbb{R}^{d \times d_k}. \tag{10}$$

The projected vectors are then fused by an element-wise Hadamard product:

$$p = \bigodot_{k=1}^{K} \tilde{x}^{(k)} \in \mathbb{R}^d. \tag{11}$$

Finally, an output projection with bias produces the pooled embedding:

$$z = W_o p + b, \quad W_o \in \mathbb{R}^{d \times d}, \; b \in \mathbb{R}^d. \tag{12}$$

This design captures higher-order modality interactions in a compact manner.

## B   EXPERIMENTAL SETUP

Following (Dong et al., 2023; Cheng et al., 2025d), for all audio samples, we conducted experiments on samples of length 65535 (approximately 4 seconds) at a sampling rate of 16 kHz. For spectrum computation, we employed a short-time Fourier transform (STFT) with a filter length of 1024, a hop length of 256, and a window size of 1024. All images were resized to $224 \times 224$ pixels. The audio model in this paper is a widely used 7-layer U-Net network with $k = 32$, generating 32 intermediate masks. All models were trained with a batch size of 128, using the Adam optimizer with parameters $\beta_1 = 0.9$, $\beta_2 = 0.999$, and $\sigma = 10^{-8}$, for 200,000 steps. Additionally, we employed warm-up and gradient clipping strategies, following (Dong et al., 2023). We compute the signal-to-distortion ratio (SDR) using museval (Stöter et al., 2018). All experiments were conducted on a single A100 GPU with 40GB display memory.

## C   SI-SDR AND SI-SDRi — IMPLEMENTATION

We evaluate separation quality per utterance in the time domain, reconstructing both mixture and estimates via iSTFT with the same analysis parameters used in training. For each utterance with $N$ sources we form 1-D waveforms for the mixture $x$, references $\{s_k\}_{k=1}^{N}$, and estimates $\{\hat{s}_k\}_{k=1}^{N}$. All signals are cropped to the common minimum length $L$ and zero-meaned prior to scoring; SI-SDR calculations run in "float64" for numerical stability. An energy guard is applied: if the absolute sum of any reference or its matched estimate is $\leq 10^{-5}$, the utterance is excluded from SI-SDR/-i aggregation and we report the number of skipped items. No temporal delay search is performed—i.e., we assume sample-level alignment from the dataset and the iSTFT pipeline.

For a single reference-estimate pair $(s, \hat{s})$ (zero-mean, length $L$), SI-SDR follows the standard scale-invariant projection:

$$\alpha = \langle \hat{s}, s \rangle / \|s\|_2^2, s_{\text{target}} = \alpha s, e = \hat{s} - s_{\text{target}}, \text{SI-SDR}(\hat{s}, s) = 10 \log_{10} \left( \|s_{\text{target}}\|_2^2 / \|e\|_2^2 \right) \tag{13}$$

No filtering beyond the scalar $\alpha$ is allowed. In the multi-source case we build the matrix $S \in \mathbb{R}^{N \times N}$ with entries $S_{k,n} = \text{SI-SDR}(\hat{s}_n, s_k)$ and obtain the permutation that maximizes the total SI-SDR via the Hungarian algorithm applied to $-S$; the resulting per-source SI-SDR values under this assignment are recorded alongside the corresponding mixture baselines.

Improvement is measured against the unaltered mixture. For each reference $s_k$ we compute $\text{SI-SDR}(x, s_k)$ using the same preprocessing, and define the sample-level SI-SDRi as the mean per-source gain under the optimal assignment $\pi^\star$:

$$\text{SI-SDR}i = \frac{1}{N} \sum_{k=1}^{N} \Big[ \text{SI-SDR}(\hat{s}_{\pi^{\star}(k)}, s_k) - \text{SI-SDR}(x, s_k) \Big]. \tag{14}$$

We also store per-source SI-SDR and mixture-baseline SI-SDR lists for analysis. Dataset-level scores are then obtained by averaging per-utterance values; we additionally report either standard deviation or a 95% bootstrap confidence interval. When breaking down by categories (e.g., query modality), we aggregate within category first and macro-average across categories.

## D  REINFORCEMENT LEARNING TRAINING DETAILS

**Policy and sampling.** The separator outputs mask proposals parameterizing a factorized Beta policy $\pi_\theta(M \mid X, Q) = \prod_{t,f,c} \text{Beta}(\alpha_{tfc}, \beta_{tfc})$ with $(\alpha, \beta) = (1 + 9m, 1 + 9(1 - m))$ from the network logits $m \in [0, 1]$. At each iteration we sample $M$ from a frozen old policy $\pi_{\theta_{\text{old}}}$ (a one-step snapshot) and reconstruct the waveform via iSTFT using the same analysis parameters as in training.

**Objective.** We maximize a PPO-style clipped surrogate with entropy regularization and an optional KL penalty:

$$\mathcal{J}_{\text{clip}}(\theta) = \mathbb{E}_{M \sim \pi_{\theta_{\text{old}}}} \Big[ \min\big(r_\theta(M)\tilde{A}, \text{clip}(r_\theta(M), 1-\epsilon, 1+\epsilon)\tilde{A}\big) + \lambda_H \, \mathcal{H}(\pi_\theta) - \lambda_{\text{KL}} \, \text{KL}(\pi_\theta \| \pi_{\theta_{\text{old}}}) \Big],$$

where $r_\theta(M) = \exp(\log \pi_\theta - \log \pi_{\theta_{\text{old}}})$ and $\tilde{A}$ is the advantage after baseline subtraction and, when enabled, group-relative normalization. We minimize $\mathcal{L}_{\text{RL}}(\theta) = -\mathcal{J}_{\text{clip}}(\theta)$ and update $\pi_{\theta_{\text{old}}} \leftarrow \pi_\theta$ after each step (single-epoch PPO).

**Advantages and baselines.** Rewards are the cosine similarities between separated audio embeddings and query-conditioned targets (audio/text/video or their mixup/adaptive variants). We use an EMA baseline $b \leftarrow \beta b + (1 - \beta) \mathbb{E}[R]$ with $\beta = 0.92$. GRPO normalization (optional) sets $\tilde{A} = (A - \mu(A))/(\sigma(A) + 10^{-6})$ within the current group.

**Default hyperparameters.** Clipping range $\epsilon = 0.2$; entropy coefficient $\lambda_H = 0.1$; KL coefficient $\lambda_{\text{KL}} \in \{0, 0.01\}$ (on by default); one Monte Carlo sample per step; mixed precision (FP16/BF16) for the separator, FP32 for reward computation; AdamW with learning rate $2 \times 10^{-4}$, weight decay 0.01; global batch size $B$ as reported in the main text; gradient clipping at 1.0; early stopping on validation reward. Unless otherwise noted, GRPO is enabled.

**Reward encoder alignment.** We apply progressive alignment of the multimodal encoder prior to RL (staged contrastive objectives with the encoder trunk largely frozen and projection heads trainable), then keep the encoder frozen during RL unless specified. This improves reward faithfulness and stability.

**Evaluation protocol.** Permutation-invariant matching is used for multi-source cases; SI-SDR/-i are computed per utterance with mixture as the improvement baseline and no delay search, following our metric appendix. All systems share identical evaluation knobs.

## E  MORE EXPERIMENTS

### E.1  ABLATION STUDIES

This section provides a systematic analysis of the key factors influencing MARS-Sep, including hyperparameters, modality fusion strategies, and encoder characteristics. Across all experiments, we observe a consistent pattern: stable policy optimization and semantically aligned reward modeling jointly determine final separation quality, and different components contribute in a complementary way.

### E.1.1 HYPERPARAMETER SETTINGS

Tables 4-7 summarize the effect of several hyperparameters in omni-queried cases. The concentration parameter $\kappa$ of the Beta policy introduces a direct trade-off between exploration and stability: small values yield overly noisy samples, while large values cause premature policy collapse. $\kappa = 9$ achieves the best balance between semantic reward optimization and signal fidelity. The PPO clip range $\epsilon$ exhibits very limited sensitivity, indicating that the trust-region update is already sufficiently stable.

In contrast, the entropy coefficient $\lambda_H$ and the KL coefficient $\lambda_{KL}$ noticeably influence exploration behavior. Moderate entropy regularization encourages continued mask diversity, while a lightweight KL penalty prevents the policy from drifting too far from its previous snapshot, which stabilizes reward estimation. Overall, these trends align with observations from GRPO-style RL training in large models—mild exploration and gentle update constraints are most effective for maintaining semantic consistency.

| $\kappa$ | 3 | 6 | 9 (Default) | 12 | 15 |
|---|---|---|---|---|---|
| Mean SDR | 6.99±0.71 | 7.84±0.74 | **7.93±0.75** | 7.74±0.75 | 7.70±0.77 |
| Mean **CLAP**$_t$ | 8.41±0.90 | 8.75±0.92 | **9.22±0.90** | 9.19±0.94 | 9.10±0.88 |

Table 4: Effect of Beta distribution concentration parameter ($\kappa$).

| $\epsilon$ | 0.01 | 0.1 | 0.2 (Default) | 0.3 | 0.4 |
|---|---|---|---|---|---|
| Mean SDR | 7.91±0.75 | 7.90±0.74 | **7.93±0.75** | 7.91±0.75 | 7.91±0.73 |
| Mean **CLAP**$_t$ | 9.00±0.95 | 9.08±0.92 | **9.22±0.90** | 9.14±0.88 | 9.12±0.88 |

Table 5: Effect of PPO clip range parameter ($\epsilon$).

| $\lambda_H$ | 0.01 | 0.1 | 0.2 (Default) | 0.3 | 0.4 |
|---|---|---|---|---|---|
| Mean SDR | 7.20±0.74 | 7.68±0.72 | **7.93±0.75** | 7.62±0.72 | 7.57±0.77 |
| Mean **CLAP**$_t$ | 8.50±0.89 | 8.76±0.93 | **9.22±0.90** | 8.79±0.92 | 8.46±1.01 |

Table 6: Effect of Entropy coefficient parameter ($\lambda_H$).

### E.1.2 MODALITY AGGREGATION MODULE

We compared several query fusion mechanisms beyond MLBP, including **Max Pooling, Average Pooling, and Learnable Weighted Sums**. Results are displayed in Table 8 and Table 9. Although Learnable Weighted Sums slightly outperform MLBP in the Audio-Query and Omni-Query settings of VGGSOUND-clean+, MLBP exhibits more stable behavior in Text-Query and Image-Query scenarios and achieves the strongest and most consistent gains on the cross-domain MUSIC-clean+ benchmark.

This advantage arises because MLBP explicitly models multiplicative interactions across modalities—especially between text and vision—allowing the reward model to form a more reliable cross-modal semantic anchor. As a result, MLBP proves particularly effective when semantic cues are complex or span multiple modalities, outperforming simpler pooling or weighting schemes under these challenging conditions.

### E.1.3 EFFECT OF PROGRESSIVE FINE-TUNING ON SOURCE DISCRIMINATION OF IMAGEBIND

To verify that **progressive fine-tuning improves ImageBind's ability to discriminate between target and non-target sounds**, for each target audio sample in the VGGSOUND test set (with its corresponding text label), we randomly selected an interference audio with a different label and constructed a mixture of the two. We then compared the cosine similarity between the target text embedding and the embeddings of both the clean target and the mixture, using the pretrained and fine-tuned ImageBind models. By averaging the similarity differences $\text{sim}(\text{emb}_{target}, \text{emb}_{mixture})$

| $\lambda_{KL}$ | 0 | 0.1 (Default) | 0.2 |
|---|---|---|---|
| Mean SDR | 7.43±0.72 | **7.93±0.75** | 7.69±0.70 |
| Mean **CLAP**$_t$ | **9.63±0.90** | 9.22±0.90 | 9.19±0.92 |

Table 7: Effect of KL Divergence coefficient parameter ($\lambda_{KL}$).

Table 8: Comparison of sound separation performance among different methods on **VGGSOUND-clean+** dataset.

| Methods | VGGSOUND-clean+ | | | | |
|---|---|---|---|---|---|
| | Mean SDR↑ | Mean SIR↑ | Mean SAR↑ | Mean SI-SDRi↑ | Mean CLAP$_t$ ↑ |
| *Text Query Sound Separation* | | | | | |
| Max Pooling | 6.83±0.68 | 8.69±1.00 | 14.19±0.75 | 4.35±0.44 | 8.99±0.92 |
| Learned Weighted Sums | 6.79±0.65 | 8.95±1.00 | **13.77±0.75** | 4.34±0.46 | 8.95±0.95 |
| Average Pooling | 6.75±0.66 | **9.20±1.00** | 13.54±0.75 | 9.18±0.79 | 8.97±0.94 |
| MLBP (ours) | **6.91±0.68** | 9.14±1.00 | 13.73±0.77 | **4.55±0.44** | **9.03±0.94** |
| *Audio Query Sound Separation* | | | | | |
| Max Pooling | 7.11±0.61 | 11.21±0.99 | 11.72±0.79 | **4.39±0.53** | 8.67±0.93 |
| Learned Weighted Sums | 7.23±0.65 | 11.57±0.99 | 11.82±0.82 | 4.24±0.55 | 8.75±0.97 |
| Average Pooling | 6.69±0.61 | **11.76±1.02** | 11.12±0.76 | 3.90±0.60 | 8.42±0.91 |
| MLBP (ours) | **7.33±0.67** | 11.63±1.00 | **12.00±0.84** | 4.36±0.50 | **8.91±0.91** |
| *Image Query Sound Separation* | | | | | |
| Max Pooling | 6.78±0.66 | 9.52±1.04 | 13.76±0.74 | 4.43±0.49 | 8.68±0.90 |
| Learned Weighted Sums | **7.15±0.69** | 10.03±1.07 | 13.95±0.75 | **4.80±0.50** | 9.04±0.92 |
| Average Pooling | 7.03±0.68 | **10.20±1.07** | 13.95±0.76 | 4.69±0.48 | 8.81±0.91 |
| MLBP (ours) | 6.93±0.67 | 10.18±1.04 | 13.41±0.72 | 4.57±0.47 | **9.19±0.91** |
| *Composed Omni-modal Query Sound Separation* | | | | | |
| Max Pooling | 7.86±0.72 | 10.31±1.00 | 14.54±0.92 | 5.13±0.45 | 8.94±0.92 |
| Learned Weighted Sums | **8.08±0.75** | 10.58±1.00 | **14.74±0.95** | **5.32±0.47** | 9.07±0.95 |
| Average Pooling | 7.87±0.74 | **10.96±1.03** | 14.52±0.91 | 5.20±0.46 | 9.18±0.94 |
| MLBP (ours) | 7.93±0.75 | 10.65±1.00 | 14.49±0.95 | 5.20±0.45 | **9.22±0.90** |

across all test samples, we obtain a robust measure of the model's ability to discriminate non-target sounds. The results in Table 10 demonstrate that the fine-tuned model consistently yields a larger average difference, confirming its improved semantic alignment in the presence of interfering sources.

### E.1.4 EFFECT OF REINFORCEMENT LEARNING AND ENCODER FINE-TUNING UNDER DIFFERENT TRAINING PIPELINES.

To disentangle the contributions of reinforcement learning (RL) and progressive fine-tuning (FT) of the ImageBind encoder, we compared four training configurations: (i) baseline supervised training with a frozen encoder, (ii) RL with frozen encoder, (iii) FT-only under supervised training, and (iv) RL combined with FT (our full model). Results on the test set in Table 11 reveal a consistent trend. The FT-only variant yields higher SAR scores but substantially lower SDR, SIR, SI-SDRi and CLAP score, indicating that the encoder becomes more sensitive to semantic cues but the conventional objective fails to enforce clean separation, leading to leakage from interfering sources. By contrast, the RL-only variant achieves improvements over the baseline across all metrics, demonstrating that policy optimization itself enhances separation fidelity even without encoder adaptation. Finally, the RL+FT variant provides the best overall performance, simultaneously improving SDR/SIR and achieving the highest SAR and CLAP scores. These findings confirm that reinforcement learning is crucial for harnessing the benefits of fine-tuned encoders while avoiding the metric trade-off observed in the FT-only setting.

This case further illustrates that while fine-tuning enhances the encoder's semantic sensitivity, reinforcement learning is indispensable to suppress residual noise and achieve clean separation. In combination, RL and FT strike a balance between semantic alignment and signal fidelity, yielding perceptually superior outputs.

Table 9: Comparison of sound separation performance among different methods on **MUSIC-clean+** dataset.

| Methods | MUSIC-clean+ | | | | |
| --- | --- | --- | --- | --- | --- |
| | Mean SDR↑ | Mean SIR↑ | Mean SAR↑ | Mean SI-SDRi↑ | Mean CLAP$_t$ ↑ |
| *Text Query Sound Separation* | | | | | |
| Max Pooling | 12.80±0.92 | 17.37±1.20 | 18.58±0.89 | 9.55±0.81 | 5.34±0.89 |
| Learned Weighted Sums | 12.60±0.82 | **17.79±1.21** | 18.23±0.84 | 9.47±0.80 | 5.39±0.92 |
| Average Pooling | 12.08±0.89 | 17.31±1.19 | 17.73±0.87 | 8.82±0.85 | 5.14±0.92 |
| MLBP (ours) | **12.91±0.93** | 17.61±1.17 | **18.28±0.93** | **9.85±0.82** | **6.18±0.93** |
| *Audio Query Sound Separation* | | | | | |
| Max Pooling | 11.13±0.87 | 18.72±1.14 | 15.56±0.84 | 7.81±1.06 | 5.38±0.96 |
| Learned Weighted Sums | 10.98±0.89 | 18.92±1.10 | 14.69±0.90 | 8.03±1.00 | **5.77±0.98** |
| Average Pooling | 10.48±0.90 | 18.69±1.03 | 14.16±0.90 | 7.62±1.08 | 5.46±1.00 |
| MLBP (ours) | **11.73±0.88** | **19.65±1.14** | **15.25±0.86** | **8.38±1.03** | 5.64±1.06 |
| *Image Query Sound Separation* | | | | | |
| Max Pooling | 13.29±1.05 | 18.69±1.20 | 18.23±1.02 | 10.35±0.95 | 6.44±1.05 |
| Learned Weighted Sums | 13.21±0.93 | 18.75±1.17 | **18.23±0.95** | 10.31±0.92 | 6.09±1.01 |
| Average Pooling | 12.80±0.96 | 18.27±1.18 | 17.85±0.97 | 9.82±0.93 | 6.36±1.03 |
| MLBP (ours) | **13.64±1.06** | **19.24±1.16** | 18.05±1.06 | **10.70±0.89** | **6.94±1.06** |
| *Composed Omni-modal Query Sound Separation* | | | | | |
| Max Pooling | 13.61±1.01 | 19.61±1.21 | 18.10±0.97 | 10.58±0.88 | 6.61±0.96 |
| Learned Weighted Sums | 13.41±0.90 | 19.54±1.18 | 17.99±0.90 | 10.43±0.81 | 6.49±0.95 |
| Average Pooling | 13.15±0.95 | 19.21±1.23 | 17.73±0.92 | 10.09±0.84 | 6.46±0.94 |
| MLBP (ours) | **13.89±0.98** | **19.90±1.18** | **17.99±0.97** | **10.78±0.81** | **6.82±0.99** |

Table 10: Average similarity differences (target - mixture) relative to the target text embedding, evaluated across the full test set. Larger values indicate stronger discrimination of non-target sources.

| Model | Avg. Difference (↑) |
| --- | --- |
| Pretrained ImageBind | $0.0035 \pm 0.0561$ |
| Fine-tuned ImageBind | $\mathbf{0.0258 \pm 0.0630}$ |

## E.2  RESOLVING SEMANTIC AMBIGUITY IN ACOUSTICALLY SIMILAR SOURCES

To qualitatively evaluate our model's ability to address the "metric dilemma", we designed a case study focused on separating acoustically similar sources. We created a challenging audio mixture containing both the sound of **tap dancing** and **typewriting** simultaneously. These sources are highly confusable as both are characterized by sharp, percussive transients with broadband spectral content, lacking the strong, sustained harmonic structures that typically aid in separation.

The task was to isolate the tap dancing using the text query "the sound of tap dancing." When this mixture was processed by the baseline OmniSep model (without RL), it achieved a high Signal-to-Distortion Ratio (SDR), yet the resulting audio was perceptually contaminated with the distinct, rhythmic clicks of the typewriter. This outcome exemplifies the metric dilemma, where a model successfully optimizes a signal-level metric while failing to achieve true semantic separation.

In contrast, our proposed MARS-Sep, guided by a multimodal semantic reward, produced a much cleaner separation. While its SDR score was marginally lower, its SIR was substantially higher, indicating superior suppression of the interfering typewriter source. To further quantify this semantic improvement, we computed the CLAP score ((Xiao et al., 2024)), defined as **the cosine similarity between the separated audio's embedding and the text query's embedding** using the CLAP model. Unlike purely signal-level metrics, the CLAP score directly measures semantic alignment across modalities, offering a more reliable indicator of whether the separated source matches the intended textual description. The comparative results are summarized in Table 12.

This case study validates that by directly optimizing for semantic consistency, MARS-Sep effectively mitigates semantic contamination and delivers perceptually superior results in scenarios where traditional signal-level metrics can be misleading. Moreover, the use of CLAP score highlights the

Table 11: Comparison of different training configurations on the VGGSOUND-clean+ test set with text queries. RL here stands for reinforcement learning and FT denotes progressive fine-tuning of ImageBind.

| Method | Mean SDR↑ | Mean SIR↑ | Mean SAR↑ | Mean SI-SDRi↑ | Mean CLAP$_t$ ↑ |
|---|---|---|---|---|---|
| Baseline (Supervised + Frozen Encoder) | 6.70±0.66 | 9.04±0.98 | 13.61±0.77 | 4.38±0.48 | 8.98±0.89 |
| RL-only (RL + Frozen Encoder) | 6.71±0.70 | 9.04±1.02 | 14.08±0.80 | 4.50±0.75 | 8.96±0.90 |
| FT-only (Supervised + Fine-tuned Encoder) | 0.75±0.64 | 1.41±1.18 | **87.13±0.15** | 0.00±0.00 | 5.48±0.95 |
| RL+FT (Full Model) | **6.91±0.68** | **9.14±1.00** | 13.73±0.77 | **4.55±0.44** | **9.03±0.94** |

Table 12: A quantitative and qualitative comparison for the "tap dancing"-"typewriting" separation task. This table presents the results for the baseline OmniSep model and our proposed MARS-Sep. The CLAP score is the cosine similarity between the separated audio embedding and the text query ("the sound of tap dancing") embedding generated by the CLAP model.

| Model | Text Query | SDR | SIR | SAR | CLAP score |
|---|---|---|---|---|---|
| CLIPSEP-NIT (Dong et al., 2023) | | 11.8540 | 24.1064 | 16.3873 | 0.3053 |
| OmniSep (Cheng et al., 2025d) | "tap dancing" | 10.8925 | **24.2579** | 17.0300 | 0.4810 |
| MARS-Sep (Ours) | | **12.0603** | 24.0055 | **17.2554** | **0.4935** |

advantage of employing cross-modal semantic evaluation, as it aligns closely with human perception of whether the separation captures the intended sound concept.

### E.3 EFFICACY OF THE PROGRESSIVE ALIGNMENT FINETUNING

To further examine the contribution of the progressive alignment strategy in our *MARS-Sep* framework, we replace the progressively fine-tuned ImageBind encoder with

(i) a frozen version without any fine-tuning (*no finetuning*),

(ii) a variant fine-tuned in a single stage on the mixed paired dataset (*1-stage finetuning*).

Table 13: Comparison of sound separation performance among different fine-tuning strategies on **VGGSOUND-clean+** dataset.

| Methods | VGGSOUND-clean+ | | | | |
|---|---|---|---|---|---|
| | Mean SDR↑ | Mean SIR↑ | Mean SAR↑ | Mean SI-SDRi↑ | Mean CLAP$_t$ ↑ |
| *Text Query Sound Separation* | | | | | |
| No fine-tuning | 6.59±0.68 | 8.82±1.01 | 13.67±0.75 | 4.23±0.46 | 8.56±0.90 |
| 1-stage fine-tuning | 6.73±0.68 | 9.24±0.99 | 13.72±0.79 | 4.40±0.46 | **9.07±0.91** |
| 3-stage fine-tuning | **6.91±0.68** | **9.14±1.00** | **13.73±0.77** | **4.55±0.44** | 9.03±0.94 |
| *Audio Query Sound Separation* | | | | | |
| No fine-tuning | 6.85±0.62 | 11.46±0.99 | 11.39±0.77 | 4.15±0.53 | 8.69±0.93 |
| 1-stage fine-tuning | 6.69±0.62 | 11.35±1.03 | 11.40±0.78 | 3.98±0.53 | 8.64±0.91 |
| 3-stage fine-tuning | **7.33±0.67** | **11.63±1.00** | **12.00±0.84** | **4.36±0.50** | **8.91±0.91** |
| *Image Query Sound Separation* | | | | | |
| No fine-tuning | **7.11±0.68** | 9.96±1.04 | **14.00±0.75** | **4.68±0.48** | 8.59±0.91 |
| 1-stage fine-tuning | 6.54±0.63 | 9.99±1.05 | 13.57±0.77 | 4.11±0.50 | 9.17±0.90 |
| 3-stage fine-tuning | 6.93±0.67 | **10.18±1.04** | 13.41±0.72 | 4.57±0.47 | **9.19±0.91** |
| *Composed Omni-modal Query Sound Separation* | | | | | |
| No fine-tuning | 7.69±0.75 | 10.34±1.01 | **14.57±0.94** | 4.98±0.48 | 9.05±0.91 |
| 1-stage fine-tuning | 7.67±0.74 | **10.70±1.03** | 14.48±0.94 | 4.84±0.50 | 8.83±0.89 |
| 3-stage fine-tuning | **7.93±0.75** | 10.65±1.00 | 14.49±0.95 | **5.20±0.45** | **9.22±0.90** |

As shown by the results in Table 13, *MARS-Sep* with progressive alignment still demonstrates clear advantages on the **VGGSOUND-clean+** dataset, though the trends differ slightly from those observed on MUSIC-clean+. For text and audio query separation, the three-stage fine-tuning strategy consistently yields the best overall signal-level metrics (SDR, SIR, SAR, and SI-SDRi), indicat-

ing that progressive, stage-wise alignment effectively enhances the multimodal encoder's domain adaptation capability. In particular, the gain in audio query separation is most evident, where the three-stage strategy surpasses both the frozen and single-stage variants across all metrics, showing its robustness in cross-modal grounding.

While the improvement from progressive fine-tuning is slightly less pronounced in the image-query setting, this can be naturally explained by the fact that ImageBind's visual encoder is pretrained on large-scale visual data and already provides a stable embedding space. Thus, image-audio alignment requires less additional adaptation than text-audio or audio-audio alignment. In contrast, the text and audio branches differ more significantly in semantic granularity and distribution, making them more sensitive to staged alignment.

Nonetheless, the three-stage strategy consistently provides the most robust performance across text, audio, and omni-modal queries, where semantic compositionality is more complex. These results confirm that progressive alignment is especially beneficial for cross-modal generalization and semantic consistency in challenging, in-the-wild mixtures.

### E.4 MEMORY CONSUMPTION AND INFERENCE LATENCY

To validate the training time and memory consumption and inference efficiency, we conduct additional experiments to measure training compute, hardware configuration, memory usage, and inference latency on standard hardware. Table 14 summarizes the results. Both OmniSep and our MARS-Sep model were trained on an Nvidia A800 40GB GPU, and the memory consumption for all modalities remains comparable at approximately 35.5 GB. In terms of training efficiency, MARS-Sep requires roughly 8 hours per epoch with a batch size of 128 and 10k steps, which is about 50% of the baseline efficiency reported by OmniSep (about 4 hours per epoch). Despite the increased training cost, our model introduces virtually no additional inference overhead: the real-time factors (RTF) across text, image, audio, and omni-modality inputs remain on par with OmniSep, with differences within normal variance. These results demonstrate that MARS-Sep maintains comparable inference-time efficiency while achieving its improvements through a modest increase in training-time computation.

| Method | Hardware | Memory Consumption | Training Time per Epoch (Batch Size=128, 10k steps) | Inference Latency per Batch(RTF) (Batch Size = 4) |
|---|---|---|---|---|
| OmniSep | $\sim$ 35.5GB | Nvidia A800 40GB | $\sim$ 4 hours | Text: 0.1283s
Image: 0.0841s
Audio: 0.0812s
Omni: 0.0801s |
| MARS-Sep (Ours) | $\sim$ 35.5GB | Nvidia A800 40GB | $\sim$ 8 hours | Text: 0.1202s
Image: 0.0816s
Audio: 0.0814s
Omni: 0.0893s |

Table 14: Efficiency reporting on standard hardware. Inference Latency is tested on VGGSOUND dataset.

### E.5 USER STUDY FOR EVALUATING CROSS-MODAL SEMANTIC ALIGNMENT

To complement the CLAP-based automatic metrics and address concerns about potential biases inherited from CLAP, we conducted a human perceptual study to directly evaluate the semantic alignment between user queries and the separated audio. We sampled 100 query-audio pairs from the test set, covering diverse sound events and including both text, image and audio queries. For each pair, we presented participants with the query and two audio excerpts-one target audio and one separated by our method or OmniSep-in randomized order.

10 non-expert listeners participated in the study, each providing 100 judgments. Participants rated the semantic consistency between the query and each audio sample on a 1-5 Likert scale, and additionally chose the sample that better matched the query in a pairwise preference setting. As shown

in Table 15, MARS-Sep achieves higher semantic-matching scores and is preferred against the target audio more often than OmniSep across all three query conditions. These results indicate that our improvements are not merely artifacts of CLAP similarities but are also perceived by human listeners, supporting the claim that MARS-Sep produces more semantically aligned separations.

| Method | Semantic Match (1-5) | Pairwise Preference (%) |
|---|---|---|
| Target Audio | $4.76 \pm 0.20$ | / |
| OmniSep | $3.42 \pm 0.51$ | 15.4 |
| MARS-Sep (Ours) | $\mathbf{3.57 \pm 0.48}$ | **23.2** |

Table 15: Human user study evaluating semantic alignment between queries and separated audio.

### E.6 ADDITIONAL QUALITATIVE RESULTS IN THE TQSS SETTING

Figure 4 illustrates representative qualitative comparisons under the TQSS scenario. For each example, we visualize the log-mel spectrograms of the mixed input, the target source, the interference source, as well as the separation outputs from the baseline method and our proposed approach. As can be observed, our method better preserves the structure of the target source while effectively suppressing interference components. More examples are available on our project webpage `https://mars-sep.github.io/`.

## F THE USE OF LARGE LANGUAGE MODELS (LLMS)

In this work, we utilized a large language model (GPT-5) for two auxiliary purposes. It was used to generate illustrative images, including those of a 'fox barking' and a 'playing congas' for Figure 2. Additionally, we leveraged its search capabilities to assist with our literature review.

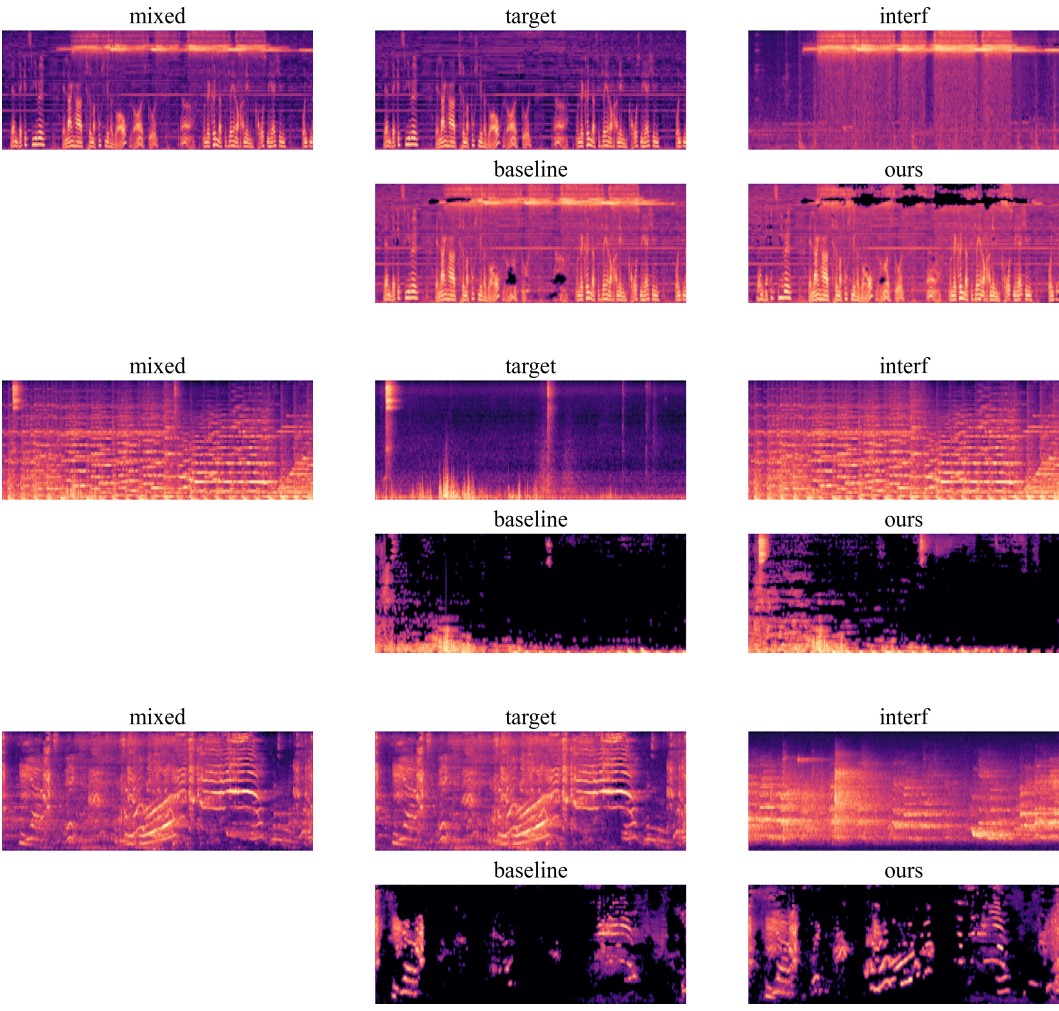

Figure 4: Qualitative comparison of separation results in the TQSS setting. Each group contains 5 spectrograms: mixed input, target source, interference source, baseline(OmniSep) separation, and our method separation.

