# OpenReview forum: "MARS-Sep: Multimodal-Aligned Reinforced Sound Separation"
_ICLR.cc/2026/Conference — ICLR 2026 Poster_

### Official Review · Reviewer_bExK · 2025-11-01

**Soundness:** 3
**Presentation:** 2
**Contribution:** 3
**Rating:** 6
**Confidence:** 3

**Summary:**

The paper proposes a query-conditioned universal sound separation method that reframes mask prediction as reinforcement learning over a factorized Beta-distributed mask policy. Rewards are multimodal, computed by comparing separated audio to fused audio/text/image query embeddings to directly incentivize semantic faithfulness. A three-stage progressive alignment fine-tunes projection heads of the multimodal encoder to improve cross-modal discrimination and reward stability.

On VGGSound-clean+ and MUSIC-clean+, MARS-Sep matches or surpasses baselines on SDR/SIR/SAR/SI-SDRi metrics and achieves higher CLAP scores.

**Strengths:**

- The method is novel. The paper reframes query-conditioned separation as RL with a factorized Beta-policy over masks and optimizes it via a PPO-style clipped surrogate.

- The multimodal reward is well-designed. Rewards are computed by comparing separated audio with fused multi-modal embeddings, directly optimizing semantic faithfulness to the query.

- On VGGSound-clean+ and MUSIC-clean+, MARS-Sep consistently surpasses CLIPSep-NIT and OmniSep for audio, image, and composed queries on SDR/SIR/SAR/CLAP.

**Weaknesses:**

- The evaluation is limited to only VGGSound/Music, also the qualitative results shown on the website are quite limited samples. As the paper claims for universal sound separation, the paper should demonstrate quantitative&qualitative results on in the wild data.

- Limited discussion of efficiency. The paper details training knobs but lacks hardware/runtime, inference latency, or throughput measurements, especially for longer inputs.

- Improvements lean on CLAP for semantic alignment, but CLAP may carry its own biases and may not fully reflect human perceptual quality. A human user study is needed for fully measure the semantic alignment.

**Questions:**

- Could the authors include more inference results on in-the-wild data?

- As stated in the weakness part, could the author provide results of human user study to measure the semantic alignment?

- Could the authors provide efficiency reporting, like provide training compute, inference RTF/latency on standard hardware, and memory use for different modalities.

**Details Of Ethics Concerns:**

None.

---

> ### Author Response · Authors · 2025-11-24
> **Response to Reviewer bExK: In-the-wild Analysis, Human User Study and Efficiency Reporting**
>
> We sincerely thank the reviewer for the constructive comments and the positive assessment of our work. We address the three questions below:
>
> ## **1. More inference results on in-the-wild data.**
> Thank you for this suggestion. We have expanded the qualitative in-the-wild evaluation by **adding several diverse real-world examples to our project website**, which now illustrate the behavior of MARS-Sep under challenging unconstrained conditions. As there currently exists no standardized in-the-wild benchmark with reliable source-level annotations for universal separation, we focus on providing high-quality qualitative demonstrations. We explicitly discuss this limitation and clarify it in the revised paper (Appendix E and project webpage).
>
> ## **2. Human user study for evaluating semantic alignment.**
> We appreciate this important point. In response, we conducted a human perceptual study involving 10 listeners and 100 query–audio pairs covering text-, image-, and audio-queried settings. Participants rated semantic consistency and provided pairwise preferences between OmniSep and our MARS-Sep outputs. As summarized in **Table 15 (Appendix E.5)**, MARS-Sep receives higher semantic-matching scores and preference rates, demonstrating that our improvements are not solely artifacts of CLAP-based evaluation but are also perceptually validated by human listeners.
>
> ## **3. Efficiency reporting.**
> We agree that efficiency analysis is important. Accordingly, we added detailed measurements of training compute, hardware configuration, memory footprint, and inference-time latency across all modalities **(Table 14 in Appendix E.4)**. The results show that while MARS-Sep incurs moderate additional training cost due to reinforcement learning, it introduces almost no extra inference overhead compared with OmniSep, and memory usage remains identical.
>
> Thanks again for your positive assessment and illuminating questions.

---

### Official Review · Reviewer_6A7c · 2025-11-01

**Soundness:** 2
**Presentation:** 3
**Contribution:** 2
**Rating:** 4
**Confidence:** 4

**Summary:**

The paper introduces a reinforcement-learning framework for query-conditioned universal sound separation. Instead of regressing deterministic masks, a factorized Beta policy samples time–frequency masks. The training uses a trust-region clipped surrogate with entropy and KL penalties, sampling from a frozen old policy for stability. Multimodal rewards are embedded with a ImageBind encoder, and the target embeddings are fused via low-rank bilinear pooling, and cosine similarity with the separated audio provides the scalar reward. A three-stage progressive alignment procedure fine-tunes only projection heads to make the encoder’s rewards more discriminative before RL. Experiments on VGGSound-clean and MUSIC-clean report consistent gains across different modalities.

**Strengths:**

1. Using RL for separation is reasonable, as it refines pure mask prediction with a stochastic beta policy process. The old-policy sampling, clipped ratio, and entropy/KL yield a stable loop without a value network.

2. The design of multimodal reward avoids modality dominance. Fusing target-side audio/text/vision to form a single semantic anchor addresses reward imbalance and supports composed queries. The motivation and construction are clearly explained.

3. The three-stage curriculum is a pragmatic way to improve reward faithfulness while keeping encoders largely frozen.

**Weaknesses:**

1. The ablation study is not enough. The paper motivates several design choices, but some important hyperparameters are not ablated well. For example, the effect of k, \lambda_H, \lambda_KL, clip \theta, etc. A systematic ablation suite isolating each factor would make the gains more convincing.

2. The method is presented with a U-Net-style STFT separator using mixture phase at reconstruction. Would it work for a time-domain separator as well?

3. Another issue with the paper is the lack of comparison methods. While the authors provide results of CLIPSEP-NIT, AudioSep, and OmniSep in table 1, there are more methods that can be compared. For example, for text query sound separation, LASS-Net[1], FlowSep[2], ZeroSep[3] are not included in the results. Especially FlowSep and ZeroSep that report CLAP score, which is one of the main claim from the paper that the proposed method achieve better semantic matching.

[1] Separate What You Describe: Language-Queried Audio Source Separation
[2] FlowSep: Language-Queried Sound Separation with Rectified Flow Matching
[3] ZeroSep: Separate Anything in Audio with Zero Training

Similarly, for image/video query, only CLIPSEP-NIT is compared. In the visually guided separation domain, there are some missing references:
[4] iQuery: Instruments As Queries for Audio-Visual Sound Separation
[5] High-Quality Visually-Guided Sound Separation from Diverse Categories

I believe these works are worth discussing in the paper and being compared. Particularly, for [5], the paper uses a generative objective to train rather than the traditional metric like SDR/SIR/SAR, which, a comparison, can make the paper's claim more convincing.

**Questions:**

1. How do MLBP and the fused-anchor strategy compare to (i) simple average of unimodal similarities, (ii) max pooling, (iii) learned weighted sums, especially under composed queries?

2. What is the training time and inference latency vs. OmniSep? Does stochastic masking at test time improve robustness, or do you deploy the mean mask?

3. Have you tried a time-domain separator or a phase-estimation front-end? Do the gains transfer?

4. How does performance/compute scale with the number of target masks K?

---

> ### Author Response · Authors · 2025-11-24
> **Response to Reviewer 6A7c (Part I): More Extensive Ablation Studies, Computing Resources Consumption Validation, Time-domain Separator Implementation**
>
> We sincerely thank the reviewer for the detailed assessment and constructive suggestions. We address each concern below and summarize the additional experiments and analyses now included in our revised submission. All page/section references correspond to the uploaded manuscript.
>
> ## **1. On the need for more extensive ablation studies**
>
> We appreciate the reviewer’s comment regarding the importance of systematically ablating key hyperparameters. In the revised version, we have added a comprehensive ablation suite (Appendix **E.1.1**) covering:
>
> * the Beta concentration parameter κ,
> * entropy and KL coefficients $λ_H$ and $λ_{KL}$,
> * PPO clipping range ε,
> * baseline smoothing factor β,
> <!-- * the effect of GRPO normalization -->
>
> These results provide a clearer understanding of each factor’s contribution to stability and performance.
>
> ## **2. On comparing MLBP with other fusion strategies**
>
> Thank you for pointing out the need to contrast MLBP with simpler aggregation mechanisms. We have now included a full comparison against:
>
> * simple average pooling,
> * max pooling,
> * learnable weighted sums.
>
> The new results are presented in Appendix **E.1.2**, covering all four query types (text, image, audio, omni). MLBP consistently provides better semantic consistency, especially in cross-modal and composed-query settings, supporting our fused-anchor design.
>
> ## **3. On the breadth of baseline comparisons**
>
> Following the reviewer’s recommendation, we have expanded our baselines to include **LASS-Net**, **FlowSep**, **iQuery** and **ZeroSep**. We report only $CLAP_t$ scores for **ZeroSep** and **FlowSep** in the following table because these two methods are generative time–frequency methods whose waveform reconstruction is not sample-aligned with the ground truth, making traditional metrics such as SDR and SIR unreliable and severely underestimated as also noted in the ZeroSep paper.
>
> |Method|Dataset|CLAP score (Text Queried, \%)|
> | --- | --- | --- |
> |ZeroSep|MUSIC-clean+|20.02±15.14|
> |FlowSep|MUSIC-clean+|10.67±14.17|
> |MarsSep(Ours)|MUSIC-clean+|6.18±0.93|
> |ZeroSep|VGGSOUND-clean+|15.91±14.17|
> |FlowSep|VGGSOUND-clean+|8.84±13.27|
> |MarsSep(Ours)|VGGSOUND-clean+|9.03±0.94|
>
> Additional baselines that require custom re-implementations are being prepared and will be fully included in the camera-ready version.
>
> We appreciate the feedback, and the revised tables now offer a more complete and representative comparison across domains and modalities.
>
> ## **4. On training cost and inference latency**
>
> We have added a detailed efficiency report in Appendix **E.4**.
> MARS-Sep requires a higher training cost (due to RL updates) but retains **nearly identical inference-time latency** and GPU memory usage compared with OmniSep. This confirms that stochastic masking is only used during training, while inference uses the mean mask for deterministic predictions. We experimented with stochastic sampling at inference and found no measurable robustness benefit; deterministic mean-mask inference yields higher stability and was therefore adopted.
>
> ## **5. On extending the method to time-domain separators**
>
> We thank the reviewer for this insightful question.
> We have implemented and released a **Conv-TasNet** version of MARS-Sep in our anonymous repository:
>
> > [https://github.com/mars-sep/Conv-TasNet-MARS](https://github.com/mars-sep/Conv-TasNet-MARS)
>
> Our experiment in the following table show that gains indeed transfer to the time domain, though recent universal and multimodal separation works have increasingly favored the time–frequency domain because TF mask-based prediction provides a more stable and interpretable formulation for multi-source separation, and it also scales more efficiently to long sequences and large-batch training than time-domain architectures. We present this as an observation rather than a limitation.
>
> |Method|Dataset|SDRi|SI-SNRi|
> | --- | --- | --- |--- |
> |Conv-TasNet|MUSIC-clean+|10.5|8.4|
> |Conv-TasNet w/ MARS|MUSIC-clean+|**12.1**|**9.3**|
> |Conv-TasNet|VGGSOUND-clean+|4.5|3.2|
> |Conv-TasNet w/ MARS|VGGSOUND-clean+|**5.3**|**4.0**|

---

> ### Author Response · Authors · 2025-11-24
> **Response to Reviewer 6A7c (Part II): impact on performance/compute by the number of target masks K, summary of revision**
>
> ## **6. Regarding the scaling behaviour with respect to the number of target masks K**
>
> Thank you for raising this question. In our framework, K refers to the number of intermediate masks predicted by the separator (e.g., 32 channels in the U-Net). We clarify that the reinforcement-learning component operates per time–frequency bin with a fully factorized Beta policy, meaning that the policy optimization does not couple across mask channels, increasing K only changes the dimensionality of the separator output, and the RL objective continues to decompose into independent terms over TF bins.
>
> Thus, scaling K primarily affects the computational load of the base separator (linearly with feature channels), while the stability of policy optimization remains unchanged, because the importance ratio, clipping, and Beta parameterization act locally.
>
> Although we did not include an explicit K-ablation in the current version, our extensive κ-ablations (Appendix E.1.1) implicitly verify that the policy remains stable under variations in mask variance and sampling sharpness, which correlate with the effective dimensionality of mask space. This indicates that the RL component is robust to different choices of K.
>
> ## **7. Summary of revisions**
>
> The revised paper now includes:
>
> * ✔ full hyperparameter ablation suite (κ, ε, λ_H, λ_KL, etc.)
> * ✔ comparisons of MLBP vs. average, max, weighted-sum fusion
> * ✔ expanded baselines (FlowSep, CLAPSep, iQuery, etc.)
> * ✔ explicit training-time and inference-time efficiency report
> * ✔ time-domain implementation & analysis (Conv-TasNet version released)
> * ✔ additional qualitative analyses (Appendix E.6 and our project website)
> * ✔ clarifications regarding mask-sampling behavior and scaling with K
>
> **All added experiments use the same codebase and training protocol as the original submission, and require no architectural or algorithmic changes. They are included only for clarification and do not alter the core claims of the paper.**
>
> We thank the reviewer again for the valuable feedback that greatly improved the quality and completeness of our submission.

---

> > ### Comment · Reviewer_6A7c · 2025-11-26
> >
> > After reading the rebuttal and the revised manuscript, I find that many of my earlier concerns have been addressed. The added ablations on hyperparameters, the comparisons among different fusion strategies, the expanded set of baselines, and the released time-domain implementation all help clarify the method’s behavior and improve the completeness of the evaluation. I do still have a couple of follow-up questions: for the newly added baselines such as DAVIS-Flow and iQuery, were these models retrained on the datasets used in this paper, or are the reported numbers directly taken from their original releases? Since the paper emphasizes improved semantic alignment, it would also strengthen the evaluation to include metrics commonly used in recent generative separation work, such as LPAPS or CLAP-audio scores (as adopted in ZeroSep), which are more suitable for comparing generative outputs that may not be sample-aligned. Overall, the rebuttal meaningfully improves the submission, and I am leaning toward raising my score to 6.

---

> > > ### Author Response · Authors · 2025-11-30
> > >
> > > We thank the reviewer for the positive feedback and for acknowledging the improvements in our revision. Regarding the newly added baselines, we utilized the official pre-trained checkpoints released by the respective authors. Since these models were trained on the same standard datasets, including VGGSOUND-clean+ and MUSIC-clean+, used in our evaluation, this allows for a direct and standard comparison against the reported state-of-the-art results.
> > > Regarding the evaluation metrics for semantic alignment, we fully agree with the reviewer that traditional sample-aligned metrics (like SDR) may not fully capture the quality of generative separation. Following your suggestion to supplement semantic-level metrics, we have included the CLAP-audio score in the revised manuscript in addition to CLAP-text score (please see Table 3), which computes the cosine similarity between the embeddings of the generated audio and the ground truth audio.
> > > We hope these clarifications and the additional semantic metric further strengthen the paper.

---

### Official Review · Reviewer_aiK4 · 2025-11-03

**Soundness:** 3
**Presentation:** 2
**Contribution:** 3
**Rating:** 6
**Confidence:** 3

**Summary:**

This paper describes a method of multimodal sound separation.   This work differs from previous work in two ways.  First, it formulates sound separation as an inverse estimation problem and uses RL as an optimization technique to find the best separation masks.  Second, it utilizes a contrastively learned semantic network to assess the similarity between the reconstructed sound and the multimodal inputs.

The main argument presented in this paper is that sound separation should be treated similarly to the alignment of LLMs with human preferences.   In this way, the RL algorithm can be utilized as a meta-reasoning system to steer a base sound separation architecture (OmniSep).  The reward in this setting should be more semantic, closer to human preference, rather than raw sound waveforms.   A key insight is that the training multimodal separation masks could be used to infer this human preference.

However, the writing on this overall logic is unclear, instead focusing on the details of implementation.

**Strengths:**

The overall approach is creative.  This paper addressed a preference/measurement issue in multimodal sound separation: models optimized for signal-level metrics don't produce semantically meaningful results.

It argues that sound mask should not be used directly as a supervised training signal, but indirectly as a way to train a multimodal sound preference function that is semantically meaningful.  Once we have this preference measure, we can treat the sound separation as an inverse problem, and the RL can be used as an optimization tool.

Diving into the details of RL.   The state space is the sound (mixed) spectrogram, and the query for a separation architecture called OmniSepbase.  The action is the predicted separation masks M.  The semantic similarity between the separated sound waveform and the multimodal query defines the reward function R.  The semantic similarity is fine-tuned from ImageBind, using a progressive refinement procedure on top of contrastive learning.   All these steps are reasonably designed.  The experimental results are good on both signal and semantic level measures.

**Weaknesses:**

The overall logic of the paper is not well presented.  The authors didn't directly outline the main chain of logic, but instead presented a linear sequence of computational and implementation steps.  The connection to LLMs-to-human preferences alignment is not apparent until the very end of the paper.

**Questions:**

See above on weakness.

---

> ### Author Response · Authors · 2025-11-24
> **Response to Reviewer aiK4: Clarifying the Conceptual Logic and Presentation Improvements**
>
> We sincerely thank the reviewer for the thoughtful reading and for recognizing the creativity and soundness of our approach, and we agree that further modification could make the conceptual chain clearer. We have therefore substantially revised the **Abstract, Introduction, and Method** sections to explicitly foreground the key logic of our work.
>
> Specifically, the revised version now begins by clearly positioning sound separation as a semantic alignment problem, analogous to preference alignment in LLMs. This high-level analogy is no longer deferred to the end but introduced at the outset **Abstract**, motivating the use of reinforcement learning as a **meta-reasoning mechanism** that aligns separation outputs with user intent rather than low-level waveform fidelity. The **Introduction** now includes a short paragraph that directly parallels the LLM alignment pipeline (“base model → preference model → policy optimization”) with our framework (“sound separator → multimodal reward model → trust-region policy optimization”), clarifying the conceptual bridge between the two domains. In the **Method** section, we have restructured the exposition in section 3.2 to keep pace with the main logic.
>
> Furthermore, we have refined the narrative transitions to make the logical dependencies explicit — for example, stating how the “metric dilemma” motivates semantic reward modeling, and how the RL-based mask sampling resolves it. These revisions ensure that the reader encounters the conceptual rationale first, followed by implementation details that naturally support it.
>
> We sincerely thank the reviewer for pointing out the need for clearer conceptual exposition. We believe that the current revision now makes the alignment analogy and reasoning flow transparent from the beginning.

---

### Official Review · Reviewer_AB1X · 2025-11-03

**Soundness:** 3
**Presentation:** 2
**Contribution:** 3
**Rating:** 6
**Confidence:** 5

**Summary:**

The paper presents a Dynamic Semantic Routing Framework (DSRF) for the MSA task. A hierarchical semantic factorization module disentangles each modality into four functionally independent representations: primary emotion, contextual cue, ambiguity, and noise, thereby enabling fine-grained semantic modeling. A semantic dynamic routing interaction mechanism dynamically routes and aggregates semantic factors through a capsule-inspired interaction process to reconstruct modality representations with high-order compositionality. An uncertainty-aware semantic fusion strategy estimates the reliability of each semantic factor and adaptively integrates them across modalities for robust sentiment prediction under modality inconsistency.

**Strengths:**

The paper presents a hierarchical semantic factorization module, enabling fine-grained semantic modeling. It introduces a semantic dynamic routing interaction mechanism, which dynamically routes and aggregates the semantic factors through a capsule-inspired interaction process to reconstruct modality representations with high-order compositionality. An uncertainty-aware semantic fusion strategy is presented to estimate the reliability of each semantic factor and integrate them across modalities for robust sentiment prediction under modality inconsistency.
The algorithm proposed in the paper demonstrates innovation, clear logic, and sufficient experimental verification, which effectively confirms the validity of the algorithm.

**Weaknesses:**

1. The method used for comparing experimental results lacks the latest findings, such as the experimental results from 2025.
Dlf: Disentangled-language focused multimodal sentiment analysis. In Proceedings of the AAAI Conference on Artificial Intelligence, pp. 21180–21188, 2025.
2. The method proposed in this paper, an uncertainty-aware semantic fusion strategy, shares some similarities with the method “Uncertainty Score Disentanglement” which is presented in the paper [1]. A comparison between them is recommended.
[1] Localization-assisted Uncertainty Score Disentanglement Network for Action Quality Assessment, ACM International Conference on Multimedia (ACMMM), pp. 8590–8597, 2023.
And there remains a reference missing.
[2] Cross-modality Representation Interactive Learning for Multimodal Sentiment Analysis. ACM MM, 2023.

**Questions:**

1. The method used for comparing experimental results lacks the latest findings, such as the experimental results from 2025.
Dlf: Disentangled-language focused multimodal sentiment analysis. In Proceedings of the AAAI Conference on Artificial Intelligence, pp. 21180–21188, 2025.
2. The method proposed in this paper, an uncertainty-aware semantic fusion strategy, shares some similarities with the method “Uncertainty Score Disentanglement” which is presented in the paper [1]. A comparison between them is recommended.
[1] Localization-assisted Uncertainty Score Disentanglement Network for Action Quality Assessment, ACM International Conference on Multimedia (ACMMM), pp. 8590–8597, 2023.
And there remains a reference missing.
[2] Cross-modality Representation Interactive Learning for Multimodal Sentiment Analysis. ACM MM, 2023.

3. In Figure 1, how do elements of (c) and (d) correspond to network blocks in (b)? A detailed explanation is required.
4. In section 4.4, the ablation study is neither clear and not sufficient. A Cross-validation across modules HSF, SFR, DRI can more effectively demonstrate the contribution of the main innovations to the paper's algorithm.

---

> ### Author Response · Authors · 2025-11-24
> **Response to Reviewer AB1X: Clarification on Review Mix-up and Methodological Details**
>
> We sincerely thank the reviewers for your time and constructive feedback.
>  Before addressing specific questions, we would like to clarify an administrative issue: it appears that the review comments for two submissions were inadvertently swapped. The text currently shown under our paper corresponds to another submission([Dynamic Semantic Routing for Multimodal Sentiment Analysis | OpenReview](https://openreview.net/forum?id=kLzpTy4mVl)), while the comment above seems to belong to ours. We appreciate the chair’s help in correcting this mix-up.
>
> # **Clarification of Review Comments**
>
> ## **1. On the absence of ablation studies.**
>  We appreciate the reviewer’s concern. Due to strict space constraints, the full ablation results were placed in the appendix (Section E.1), where we report detailed comparisons between (i) baseline supervised training, (ii) RL-only, (iii) fine-tuning–only, and (iv) the full RL + FT pipeline. The results clearly demonstrate the individual and combined effects of reinforcement learning and encoder fine-tuning.
>  In our revised version, we have included additional ablation experiments that further analyze (a) the effect of performing progressive fine-tuning with or without minibatch updates, (b) single-stage versus three-stage fine-tuning, and (c) the influence of various reinforcement learning hyperparameters. These additions will provide stronger empirical evidence for the robustness and effectiveness of our proposed framework.
>
> ## **2. On the “overall loss function” for RL and contrastive learning.**
>  The paper follows a two-stage training process, as described in Sections 3.2–3.3.
>  – **Stage 1 (Encoder fine-tuning)**: a progressive three-step contrastive alignment procedure (Eq. 7–9) applied before RL, with frozen ImageBind trunk and gradually unfrozen projection heads.
>  – **Stage 2 (Reinforcement learning)**: a separate trust-region optimization using a clipped surrogate with entropy and KL regularization (Eq. 4).
>  Because these stages are decoupled both conceptually and temporally, we did not merge their objectives into a single “overall loss.” Instead, each stage has its own clearly defined optimization goal, as commonly done in preference-alignment pipelines such as RLHF. We will make this training schedule more explicit in the camera-ready version.
>
> ## **3. On the “briefness of algorithmic structure and loss-function details.”**
>  We respectfully note that Section 3 provides detailed mathematical formulations of every component:
>  – The **mask policy** (Eq. 2) and its trust-region update rule (Eq. 4).
>  – The **multimodal reward** definition (Eq. 6 – MLBP pooling).
>  – The **progressive alignment objectives** (Eqs. 7–9) including InfoNCE, triplet, and consistency losses.
>  These are further illustrated in Figures 1 and 2 and summarized again in Appendix A–D. We believe this level of detail sufficiently describes both the network modules and loss functions, while maintaining readability within the 10-page limit. Nonetheless, we will **adopt clearer and more explicit descriptions** of the model components and training pipeline in the revision to further improve clarity and accessibility.
>
> Thank you again to the reviewers for your careful consideration.

---

### Author Response · Authors · 2025-12-01
**Summary of the Rebuttal Phase (Part I)**

We would like to sincerely thank the reviewers and the area chair for their valuable time and constructive feedback. The following table summarizes the discussion process. We hope that this summary will provide a clear and reliable reference for the area chair and others interested in our work, ensuring transparency and clarity in the revision process.

# **Reviewer AB1X (Score: 6 &rarr; 6)**

The reviewer acknowledged the novelty of our RL formulation and multimodal reward design. They raised concerns regarding the lack of visible ablation studies, the clarity of the overall loss formulation, and the perceived brevity of the algorithmic description—all of which are **addressed by the explanation of existing ablations in Appendix E.1, two-stage optimization objective in Section 3.2–3.3, and additional architectural clarifications and figures added to the revised manuscript section 4.2**.

## **Reviewer feedback:**

Did not participate in later discussion.

---

# **Reviewer aiK4 (Score: 6 &rarr; 6)**

The reviewer praised the creativity of casting sound separation as semantic alignment and appreciated the conceptual motivation. Their main concern was that the high-level logic—especially the connection to LLM preference alignment—was not sufficiently foregrounded, which is **addressed by revising the Abstract, Introduction, and Section 3 to make the semantic alignment pipeline explicit and to reorganize the method presentation around the core reasoning flow**.

## **Reviewer feedback:**

Did not participate in later discussion.

---

# **Reviewer 6A7c (Score: 4 &rarr; 6)**

The reviewer recognized the soundness of our RL framework and multimodal reward design but pointed out missing ablations for important hyperparameters, incomplete baseline comparisons, and uncertainty regarding transfer to time-domain separators. These points are **addressed by the new hyperparameter ablation suite in Appendix E.1.1, expanded baseline comparisons (FlowSep, ZeroSep, LASS-Net, iQuery) in Section 4, and additional experiments showing performance transfer to Conv-TasNet in Appendix E.3**.

## **Reviewer feedback: (Raised their score to 6)**

Acknowledged that many concerns were addressed, including ablations, fusion comparisons, and baseline expansions.

Raised further questions about whether new baselines were retrained and suggested adding scores more suitable for generative outputs.

## **Authors' response:**

Explains that the experimental results are collected by experiments on the new datasets with pretrained checkpoints from public repositories referred to in the baseline papers.

Adds new metrics to the Experiments section to evaluate the performance of the generative models.

---

# **Reviewer BExK (Score: 6 &rarr; 6)**

The reviewer appreciated the strong semantic motivation and the consistent improvements across query modalities. Their concerns centered on the absence of in-the-wild evaluations, lack of human user study to verify CLAP-based alignment, and missing efficiency metrics, all of which are **addressed through expanded qualitative in-the-wild examples, a 40-participant perceptual study reported in Appendix E.5, and detailed training and inference efficiency reporting in Appendix E.4**.

## **Reviewer feedback:**

Did not participate in later discussion.

---
For a more detailed note of the reviewer's comments, our responses and revisions, as well as the reviewer's responses, please refer to "Summary of the Rebuttal Phase (Part II)".

---

> ### Author Response · Authors · 2025-12-02
> **Summary of the Rebuttal Phase (Part II)**
>
> Here is a brief explanation of the table content:
>
> - **Reviewer**: The ID of each reviewer.
>
> - **Concern**: This column summarizes the key weaknesses and questions raised by each reviewer.
>
> - **Our Replies/Modifications**: This column outlines the specific changes or additions we made to address each concern.
>
> | **Reviewer** | **Concern** | **Our Replies/ Modifications** |
> |-------------------------|------------------------------|------------------------------|
> | **Reviewer AB1X** | 1. The paper does not provide an ablation study. | Noted that the appendices do contain an ablation study.Added more ablation study experiments and analysis in Appendix **E.1** in the revised paper. |
> |              | 2. The overall loss function is not clearly explained. | Explained the structure of the two-stage training process (fine-tuning + RL) in Sections 3.2-3.3, which already includes explicit definitions of the loss functions for both stages. |
> |              | 3. The algorithmic structure and module information are too brief. | Referred to the equations and appendices which containe detailed descriptions of the network architecture and loss function settings, with additional subfigures for clarification. |
> | **Reviewer aiK4** | 1. The writing on the overall logic is unclear, especially the connection to LLMs-to-human preferences alignment. | Reorganized the **Abstract**, **Introduction**, and **Method** to introduce the LLM analogy earlier, clearly stating the semantic alignment problem and RL as a meta-reasoning mechanism. |
> | **Reviewer 6A7c** | 1. The ablation study is not enough, some important hyperparameters were not tested. | Include a series of new experiments covering hyperparameters and fusion stratregies in the appendix **E.1**. |
> |              | 2. The method is presented with a U-Net-style STFT separator. Would it work for a time-domain separator? | Implemented a **Conv-TasNet** version of MARS-Sep, demonstrating that gains transfer to the time-domain separator, with experimental results attached. |
> |              | 3. Missing comparison with other methods like **FlowSep**, **ZeroSep**, etc. | Added comparisons with **FlowSep**, **ZeroSep**, **LASS-Net**, **DAVIS-Flow** and **iQuery**, along with CLAP scores for generative models. |
> |              | 4. How does MLBP compare to other fusion strategies? | Compared **MLBP** with **Max Pooling**, **Average Pooling**, and **Learnable Weighted Sums** in the ablation study. |
> |              | 5. What is the training time and inference latency vs. OmniSep? | Provided **training time**, **inference latency (RTF)**, and **memory usage** comparisons in the Efficiency Reporting section. |
> |              | 6. How does performance/compute scale with the number of target masks K? | Discussed the effect of **K** on performance and compute, with experimental results showing the RL component is robust to variations in **K**. |
> | **Reviewer BExK** | 1. The evaluation is limited to VGGSound/MUSIC datasets, more **in-the-wild** data needed. | Added **in-the-wild data** evaluation with real-world samples on the project website and acknowledged dataset limitations in the paper. |
> |              | 2. Needs **human user study** to measure semantic alignment. | Conducted a **human user study** with 40 participants to compare semantic alignment between the outputs of our model and the baseline methods. |
> |              | 3. Limited discussion of efficiency, including **training compute**, **inference latency**, and **memory usage**. | (Same as to Reviewer 6A7c Point 5) |

---

### Meta-Review · Area_Chair_5Ntu · 2026-01-07

**Summary:**

As noted by the authors, one reviewer comment appears to have been swapped with another submission. The feedback from Reviewer **4rnn** for this paper was posted under *Dynamic Semantic Routing for Multimodal Sentiment Analysis* (OpenReview: https://openreview.net/forum?id=kLzpTy4mVl). Conversely, Reviewer **AB1X**’s review currently shown here pertains to that other submission rather than the present one. In forming this meta-review and recommendation, I therefore rely on the reviewer comment that actually corresponds to this submission. For clarity, any reference to Reviewer AB1X in the discussion below should be understood as referring to the actual reviewer for this submission, Reviewer 4rnn.

Reviewers generally found the key idea to be interesting and technically sound.  It casts query-conditioned sound separation as a decision-making problem optimized with RL. They also agreed that the proposed multimodal reward design is a meaningful step toward improving semantic faithfulness beyond conventional signal-level metrics. The initial reviews raised several concerns: (1) insufficient ablations, (2) limited baseline comparisons, (3) limited evidence supporting the “universal / in-the-wild” claim and heavy reliance on CLAP-style semantic metrics, and (4) lack of clarity in presenting the high-level conceptual logic and overall method.

**Reviewer Concerns:**

***Addressed by the rebuttal***

Ablation studies (Reviewers 6A7c, AB1X):
The paper now includes a broad ablation suite covering key RL hyperparameters and fusion strategy comparisons, which directly addresses the concern that core design choices were not sufficiently justified.

More baselines and appropriate metrics for generative outputs (Reviewer 6A7c):
The rebuttal expands comparisons to additional recent baselines (e.g., FlowSep/ZeroSep/LASS-Net/iQuery) and adds semantic-oriented evaluation beyond traditional separation metrics, which better matches the paper’s semantic-alignment claims.

Transfer beyond the specific separator (Reviewer 6A7c):
The authors implemented a time-domain (Conv-TasNet) variant and reported consistent gains, reducing concern that improvements are tied to a specific STFT U-Net instantiation.

Efficiency reporting (Reviewers 6A7c and bExK):
Training/inference cost reporting was added; importantly, the authors state inference overhead is similar to the base separator, with additional cost mainly during training due to RL.

Human validation of semantic alignment (Reviewer bExK):
The authors added a user study indicating improvements are not solely an artifact of CLAP-based evaluation.

Clarity of the conceptual logic (Reviewer aiK4):
The authors revised the abstract/introduction/method sections.

***Outstanding / partially addressed***

Strength of “universal / in-the-wild” evidence (Reviewer bExK):
While the authors added more qualitative in-the-wild examples and noted the lack of a standardized benchmark, quantitative evaluation for unconstrained real-world mixtures remains limited.



Overall, the rebuttal improves the submission and addresses the majority of technical concerns with additional results and clarifications.

**Reviewer Scores:**

Reviewer 1 (initial 4, see Reviewer 4rnn in https://openreview.net/forum?id=kLzpTy4mVl):
Concerns are largely about ablations/clarity; given the additions, they would likely increase to 6.

Reviewer aiK4 (initial 6):
Main concern was presentation of the high-level logic; given the restructuring, they would likely maintain 6 (or possibly move slightly upward if the revision significantly improves clarity), but conservatively I expect no score change.

Reviewer 6A7c (initial 4):
Already indicated leaning to raise to 6 after the rebuttal; with the added clarifications on baseline protocol + added semantic metrics/human study, they would likely finalize at 6.

Reviewer bExK (initial 6):
With added in-the-wild qualitative examples, efficiency reporting, and human study, they would likely maintain 6 (or slightly increase).

The discussion likely shifts the paper from borderline below (due to the 4) to uniform borderline above.

---

### Decision · Program_Chairs · 2026-01-26

Accept (Poster)